# Integrating Crop Modeling and Machine Learning for the Improved Prediction of Dryland Wheat Yield

**Zhiyang Li [1], Zhigang Nie [1,\*] and Guang Li [2]**

[1]  College of Information Science and Technology, Gansu Agricultural University, Lanzhou 730070, China; lizy@st.gsau.edu.cn

[2]  College of Forestry, Gansu Agricultural University, Lanzhou 730070, China; lig@gsau.edu.cn

**\***  Correspondence: niezg@gsau.edu.cn

**Abstract:** One of the crucial research areas in agricultural decision-making processes is crop yield prediction. This study leverages the advantages of hybrid models to address the complex interplay of genetic, environmental, and management factors to achieve more accurate crop yield forecasts. Therefore, this study used the data of wheat growth environment, crop management, and historical yield in experimental fields in Anding District, Dingxi City, Gansu Province from 1984 to 2021 to construct eight machine learning models and ensemble models. Furthermore, Agricultural Production Systems sIMulator (APSIM), machine learning (ML), and APSIM combined with machine learning (APSIM-ML) were employed to predict wheat yields in 2012, 2016, and 2021. The results show that the APSIM-ML weighted ensemble prediction model, optimized to minimize the MSE, performed the best. Compared to the optimized ML and APSIM models, the average improvements in the RMSE, RRMSE, and MBE for the test years were 43.54 kg/ha, 3.55%, and 15.54 kg/ha, and 93.96 kg/ha, 7.55%, and 104.21 kg/ha, respectively. At the same time, we found that the dynamic flow of water and nitrogen between the soil and crops had the greatest impact on wheat yield prediction. This study improved the accuracy of dryland wheat yield prediction in Gansu Province and provides technical support for the intelligent production of dryland wheat in the loess hilly area.

**Keywords:** model algorithm construction; prediction of dryland wheat yield; APSIM; ML; hybrid weighted ensemble model

## 1. Introduction

Crop yield prediction stands as a pivotal research domain within agricultural decision-making processes. Presently, the methods used for forecasting the dryland wheat yield in Gansu primarily revolve around crop and statistical models. These individual models leverage data on environmental and management factors to make predictions, exhibiting a certain level of accuracy. However, the continual integration of new technologies, data, data-processing methods, and more efficient computational techniques with simulated data from crop models demands exploration. This pursuit aims to design predictive models that are not only more accurate and robust but also possess a stronger interpretability than the existing models. Such advancements are crucial for their application in crop production decision-making practices. The Agricultural Production Systems sIMulator (APSIM) combined with machine learning (ML) algorithms, which we have designed, is based on this premise and serves as a step forward in this area of research.

Crop models express the plant growth process through mathematical equations on computers, aiding scholars in summarizing and correlating complex crop growth phenomena. They allow an in-depth understanding of agricultural system operations and predict crop yields, as well as forecasting the impact of environmental shifts on crops [1–3]. In recent years, crop models have achieved significant research milestones in predicting crop growth and development [4], variety selection and optimizing breeding [5], water and nutrient management [6], and climate adaptation assessment [7]. While single-process-based

crop models comprehensively capture crop growth timing, frequency, and intensity, they exhibit limitations. These limitations involve the excessive simplification or ambiguous descriptions of specific procedures and uncertainties in parameterization, subsequently affecting the accuracy of prediction outcomes in crop yield forecasting [8,9]. This article summarizes key studies in agricultural research that utilize crop models, focusing on aspects such as crop yield, drought risk, and optimal fertilization strategies (Table 1).

**Table 1.** The summary outlines the research references, the crop models used, and the main findings of each study.

| Study | Model | Key Findings | Reference |
|---|---|---|---|
| Islam et al. | DSSAT (rice) | The BRRI dhan 29 variety showed the best performance under a nitrogen fertilizer treatment of 150 kg/ha, yielding 6000 kg/ha. | [10] |
| Kipkulei et al. | DSSAT-CERES (maize) | In Trans Nzoia County, Kenya, the KH600-23A maize variety outperformed H614 by 7.1% under the optimal nitrogen application strategy of 100 kg/ha in mid-February. | [11] |
| Zhao et al. | APSIM-Wheat | A strong positive correlation ($R^2$: 0.90–0.97) was found between the predicted and observed wheat growth durations in the middle and lower reaches of the Yangtze River Plain, indicating the high predictive accuracy of the model. | [12] |
| Wang et al. | APSIM (maize) | The APSIM model assessment of drought risk in Liaoning showed that spring maize in the Chaoyang and Fuxin districts faces the highest drought risk. | [13] |
| Morel et al. | APSIM (annual crops) | In Sweden's high latitude areas, a temperature increase of 1 °C maximizes barley and oats yields, while a 2–3 °C increase optimizes silage maize production. | [14] |
| Vogeler et al. | APSIM (runoff) | The study highlighted the significant impact of rainfall intensity and surface conductivity on water and nitrogen runoff loss in poorly drained clay soils. | [15] |
| Kumar et al. | APSIM (wheat phenology) | High accuracy in simulating winter wheat phenological stages in Nordic regions was achieved with an $r^2$ of 0.97 and RMSE of 3.98–4.15 on the BBCH scale. | [16] |

On the other hand, machine learning, as a pivotal technology in AI, when applied across the entire agricultural value chain, can enhance the production efficiency of the entire chain [17]. Machine learning techniques have shown promising performance in numerous applications in smart agriculture and have become one of the primary technical means for future research in smart agriculture-related issues [18,19]. The distinction between ML and crop models in forecasting crop yields lies in its more data-driven nature. Machine learning can handle non-linear relationships and exhibits adaptability, generalization, and feature learning and selection capabilities, thereby enhancing the prediction accuracy and flexibility [20]. Machine learning models, by training on historical data, automatically learn and extract relationships between features, constructing a mathematical function or model that maps input features to the target variable, namely the crop yield [21,22]. This can be achieved through supervised learning algorithms, where the model is trained based on known input features and their corresponding target variables, allowing for a more flexible modeling of complex relationships and thus more accurate crop yield predictions [23]. Machine learning methods range from basic regression models to sophisticated deep

learning (DL) algorithms [24]. We have listed some applications of machine learning in crop yield prediction (Table 2).

**Table 2.** Summary of recent studies on crop yield prediction using machine learning models.

| Study | ML Model | Key Findings | Reference |
|---|---|---|---|
| Yamparla et al. | Various (Gradient Boosting, Linear Regression, etc.) | Random forest achieved 95% accuracy in predicting crop yields in India, standing out among various machine learning techniques based on extensive historical and environmental data. | [25] |
| Burdett and Wellen | Random Forest, Decision Tree | In southwestern Ontario, Random Forest and Decision Tree models showed superior yield prediction for corn and soybean, analyzing soil attributes across 17 fields. | [26] |
| Paudel et al. | Machine Learning-based Method | A machine learning method for regional crop yield prediction in Europe was introduced, outperforming traditional models with lower errors, especially effectively close to the harvesting time. | [27] |
| Chergui | Deep Neural Network, Random Forest | Enhanced wheat yield predictions in two Algerian provinces through data augmentation, with a Deep Neural Network leading in one (RMSE of 4 kg/ha) and Random Forest in another (RMSE of 5 kg/ha). | [28] |
| Recently, Neural Network-based crop yield prediction models have also demonstrated exceptional accuracy [29–34]. | | | |
| Srivastava et al. | Convolutional Neural Network (CNN) | A CNN model for winter wheat yield prediction in Germany reduced the RMSE and MAE significantly, showing high accuracy over other machine learning models in a 20-year dataset. | [35] |
| Khaki et al. | Residual Neural Network (CNN and RNN) | A Residual Neural Network was employed for corn and soybean yield prediction in the U.S., revealing critical insights into temporal variations through environmental and operational data. | [36] |
| Gavahi et al. | Deep Yield (ConvLSTM with 3DCNN) | The Deep Yield model, using ConvLSTM and 3DCNN and trained on historical soybean data and MODIS imagery, outshone traditional models in predictive accuracy across the U.S. | [37] |

Using machine learning methods with heuristic self-learning features can enhance the understanding and prediction of critical stages in crop growth, such as flowering, fruition, maturation, and harvesting [38]. The accuracy of machine learning algorithms largely depends on a substantial amount of training data, while crop models can generate extensive dynamic data on crop growth conditions, such as daily changes in crop biomass, photosynthetic rates, and water use efficiency. Moreover, based on the integrated model proposed by Balakrishnan and Govindarajan Muthukumarasamy [39], as well as the approach suggested by Zou et al. [40] and Paudel et al. [41], which combines crop simulation with machine learning models for crop yield prediction, these reference cases indicate that integrated models can better simulate crop yield predictions. Therefore, this study conducts simulated predictions of dryland wheat yields based on a comprehensive framework of crop modeling (APSIM) and machine learning (ML). The research aims to effectively utilize a substantial amount of process data generated by APSIM simulations, along with the existing soil, meteorological, and management data, as part of the features for machine learning inputs. This approach aims to improve the accuracy of simulating and predicting wheat yields in Gansu Province, providing a theoretical foundation for the precise forecasting of dryland wheat yields and supporting agricultural intelligent production.

## 2. Materials and Methods

The methodology of this experiment involved collecting daily weather features, soil characteristics, environmental factors, and daily output data from APSIM simulations at the experimental fields. These data were utilized as input features for the machine learning (ML) model. The simulation process of daily wheat yield using the APSIM output served as the basis, where the actual yearly yields were incorporated to reconstruct the daily wheat yield data more closely resembling the real yield on a daily basis. This reconstructed data served as the target variable for training the machine learning prediction model ("Section 3.1"). Finally, the model's performance was evaluated by contrasting the predicted yield data with the actual yield data, taking into account wheat's phenological stages from heading to full maturity. The conceptual framework of this research experiment is depicted in Figure 1.

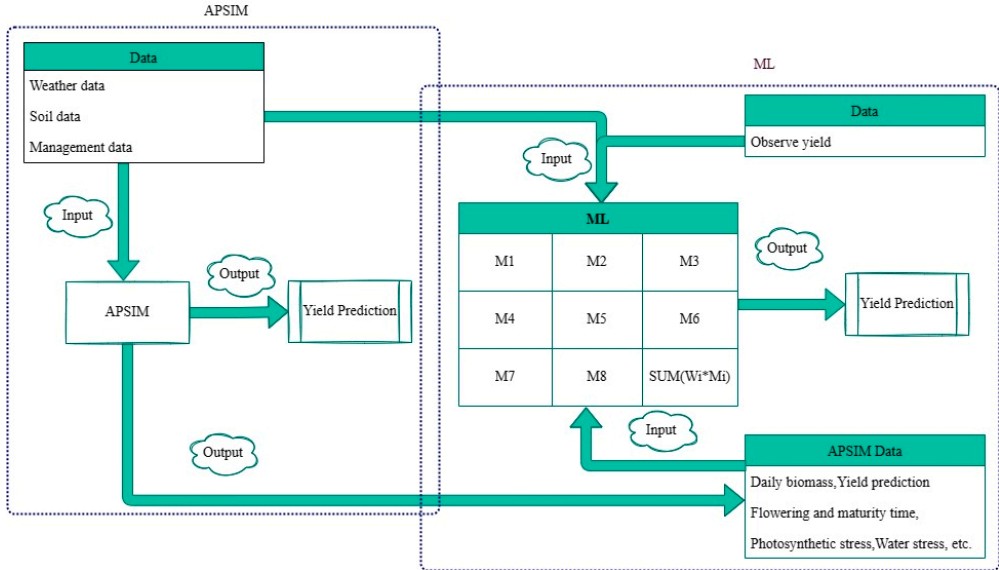

**Figure 1.** Conceptual framework of this research experiment. It comprises two major modules: APSIM and ML. Here, Mi represents different ML prediction models, Wi denotes the varying weights assigned to different ML prediction models, and SUM(Wi*Mi) signifies the ensemble prediction model.

### 2.1. Overview of the Research Area

The experimental site is located in the southern–central part of Gansu Province, at an altitude of 2000 m (Figure 2). The climate falls within the temperate semi-arid zone. The area receives an annual average solar radiation of 592.9 (kJ/m$^2$), with 2476.6 h of sunshine annually and an average temperature of 6.4°C. Additionally, the accumulated temperature $\geq 0\ °C$ averages 2933.5 °C annually, and the accumulated temperature $\geq 10\ °C$ averages 2239.1 °C annually, with a frost-free period of 140 days. The soil type in the experimental area is classified as loess soil, characterized by a bulk density of 1.19 (g/cm$^3$), a pH value of 8.36, organic matter content of 12.01 (g/kg), total nitrogen content of 0.76 (g/kg), and total phosphorus content of 1.77 (g/kg). The region is predominantly flat with no irrigation facilities. The average annual rainfall is 391.1 mm (1984–2021), but it is low, erratic, and exhibits significant inter-annual variability with a coefficient of variation of 18.5%, frequently experiencing droughts. Moreover, more than 50% of the precipitation occurs during the crop fallow period (late summer and autumn). The annual evaporation reaches 1531 mm, and the precipitation guaranteed by 80% probability is 365 mm. This area typifies semi-arid rain-fed agriculture in China, following a one-crop-per-year system. Spring wheat is extensively cultivated in this loess hilly–gully region.

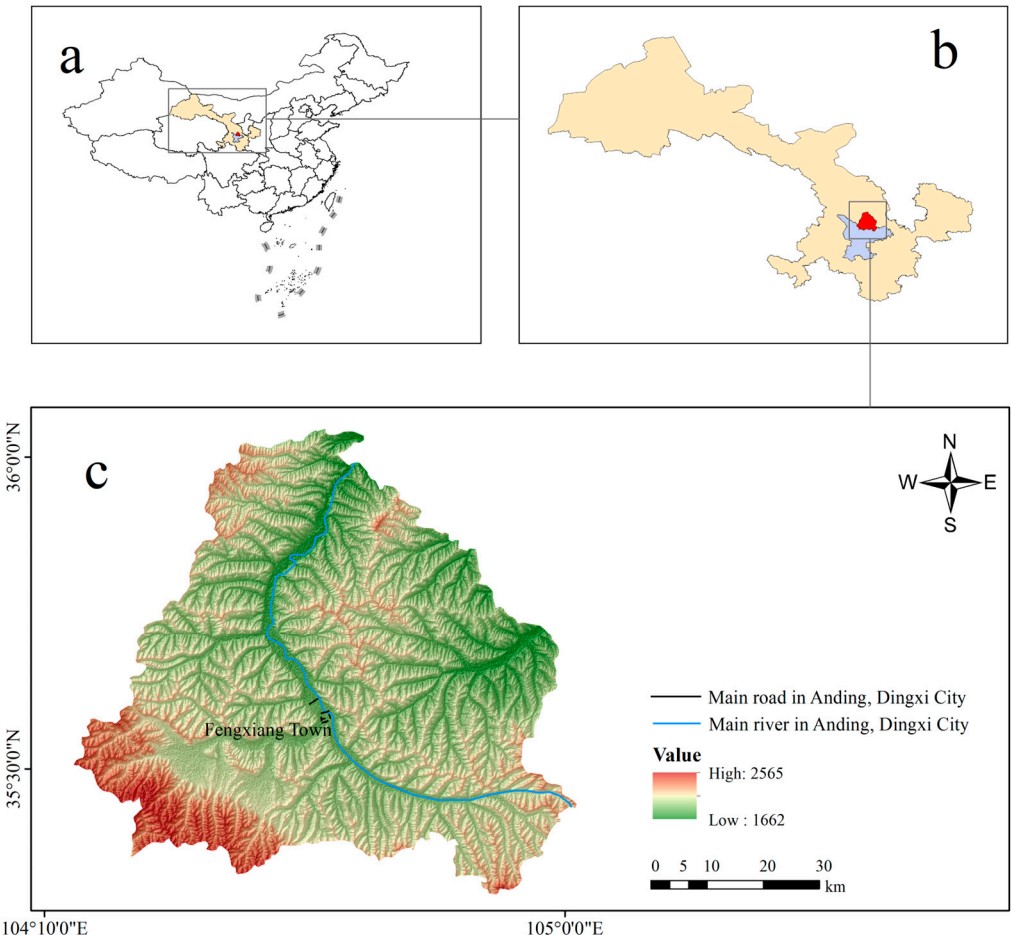

**Figure 2.** Location map of the study area. The study area, Fengxiang Town, is located at point (**c**), at the intersection of the major river and the main roads in Anding District, Dingxi City. The (**a**) is China, (**b**) is Gansu Province, (**c**) is Anding District, Dingxi City.

The selected indicator crop for the experiment was the local spring wheat variety "Ganchun 32". The sowing rate was 18,750 (kg/ha), with a seed spacing of 0.20 m. Each treatment uniformly applied fertilizers and field management practices based on local experience, incorporating 42,000 (kg/ha) of calcium superphosphate and 22,000 (kg/ha) of urea as the basal fertilizer. Prior to sowing, these two types of fertilizers were uniformly spread and incorporated into the soil by plowing to a depth of 0.20–0.30 m. Manual harvesting was conducted at the maturity stage of the spring wheat. The experiment was arranged in a randomized complete block design with three replications.

*2.2. Soil, Meteorological, Crop Management, and Yield Data Inputs to the APSIM and ML Models*

2.2.1. Soil Data

The soil data used in the simulation were obtained by the precise measurements of soil samples collected from the experimental field and analyzed in the laboratory. Thirteen soil characteristics were considered across nine different soil profile depths. These properties encompass soil bulk density, pH levels, organic matter content, total nitrogen, total phosphorus, sand content, clay content, air-dried moisture content, wilting coefficient, field capacity, saturation capacity, available water content, lower limit of available water, and soil hydraulic conductivity. The soil data in Table 1 represent the average of multiple measurements taken from 2015 to 2021 (Table 3).

**Table 3.** Soil property parameters in the experimental area.

| Parameter | Soil Depth (mm) | | | | | | | | |
|---|---|---|---|---|---|---|---|---|---|
| | 0–50 | 50–100 | 100–300 | 300–500 | 500–800 | 800–1100 | 1100–1400 | 1400–1700 | 1700–2000 |
| Bulk density (g/cm$^3$) | 1.29 | 1.23 | 1.33 | 1.20 | 1.14 | 1.14 | 1.25 | 1.12 | 1.11 |
| pH (1:5 water) | 8.320 | 8.370 | 8.330 | 8.300 | 8.320 | 8.370 | 8.420 | 8.430 | 8.400 |
| Air-dried moisture (mm/mm) | 0.01 | 0.01 | 0.05 | 0.07 | 0.09 | 0.10 | 0.11 | 0.12 | 0.13 |
| Wilting coefficient (mm/mm) | 0.09 | 0.09 | 0.09 | 0.09 | 0.09 | 0.11 | 0.11 | 0.12 | 0.13 |
| Field capacity (mm/mm) | 0.27 | 0.27 | 0.27 | 0.27 | 0.26 | 0.27 | 0.26 | 0.26 | 0.26 |
| Saturated moisture (mm/mm) | 0.46 | 0.49 | 0.45 | 0.50 | 0.52 | 0.52 | 0.48 | 0.53 | 0.53 |
| Lower available moisture (mm/mm) | 0.09 | 0.09 | 0.09 | 0.09 | 0.10 | 0.12 | 0.13 | 0.18 | 0.22 |
| Soil water conductivity (mm/h) | 0.60 | 0.60 | 0.60 | 0.60 | 0.60 | 0.60 | 0.60 | 0.60 | 0.60 |

### 2.2.2. Meteorological Data

The meteorological data were acquired from the Gansu Provincial Meteorological Bureau's website (cma.gov.cn, accessed on 10 March 2023). Daily weather data from Anjiaopo Village, Fengxiang Town, Anding District, Dingxi City (spanning from 1984 to 2021) includeed parameters such as day length (day), solar radiation (radn), maximum temperature (maxt), minimum temperature (mint), precipitation (rain), and vapor pressure (vp). These data were utilized for the APSIM simulation model and training the ML prediction model.

### 2.2.3. Crop Management Data

The planting density and fertilization methods were standardized based on the field management experience at the experimental site, and detailed records were maintained. Planting and harvesting dates were determined based on the local field management experience, taking into account the annual climate variations in Dingxi City, Gansu Province, and were recorded accordingly.

### 2.2.4. Crop Yield Data

The crop yield data used consisted of two parts: yield data from 1984 to 2014 were obtained from the Statistical Yearbook of Gansu Province published by the Gansu Provincial Bureau of Statistics (gansu.gov.cn, accessed on 10 March 2023), while yield data from 2015 to 2021 were collected through actual measurements in the experimental fields.

### 2.3. APSIM Model-Simulated Data Inputs to the ML Models

To operate the APSIM in the typical agricultural area of the loess hilly region, we employed field data from Anjiapo Village, Fengxiang Town, Anding District, Dingxi City, Gansu Province for the APSIM's utilization. This was conducted to optimize the model parameters, update and refine the model's localized standard parameter database, and enhance the model's adaptation to the local conditions for wheat yield formation. Following parameter optimization, we integrated field trial data from 1984 to 2021, meteorological records, and statistical yearbook information to compare and analyze the error between the

actual and simulated yield values before and after algorithmic improvements, validating the APSIM-Wheat model. During the experimental period from 1984 to 2021, the predicted results of the APSIM model based on meteorological data, soil data, field management practices, and other data were highly correlated with the actual yield data ("Section 3.1"). The daily output data generated from this APSIM experiment were used as input for training the ML model.

The APSIM's daily output data served as input for the ML model. The initial step in integrating the datasets required for training and testing the model involved extracting all simulated output data from the APSIM. Subsequently, these acquired APSIM-simulated output data were prepared to be merged into the initial dataset to form a new comprehensive dataset. Following the process outlined in "Section 2.4.5", 28 distinct variables from the APSIM's output (refer to Table 4 for details) were chosen as input features for the ML model.

**Table 4.** APSIM model-simulated variables considered as input features in the ML models.

| Acronym | Description |
| --- | --- |
| day_of_year | Day of year (day) |
| Yield | Crop yield (kg/ha) |
| Biomass | Crop biomass (kg/ha) |
| root_depth | Depth of roots (mm) |
| flowering_date | Day number of flowering (doy) |
| maturity_date | Day number of maturity (doy) |
| Lai | Leaf area index ($m^2/m^2$) |
| Ep | Plant water uptake (mm) |
| Es | Evaporation from soil (mm) |
| N_demanded | N demand of plant ($g/m^2$) |
| grain_n_demand | N demand of grain ($g/m^2$) |
| grain_wt | Weight of grain ($g/m^2$) |
| Grain1GrainTotalN | GrainTotal Grain1 nitrogen ($g/m^2$) |
| sw_stress_photo | Soil water stress for photosynthesis (0–1) |
| sw_stress_expan | Soil water stress for leaf expansion (0–1) |
| water_table | Water table depth (mm) |
| Runoff | Runoff (mm) |
| dlt_n_min_tot | Humic N mineralization (kg/ha) |
| Nitrification | Nitrogen moved by nitrification (kg/ha) |
| dlt_n_min | Het N mineralization (kg/ha) |
| Sws | Soil water (mm/mm) |
| sw_supply | Soil water supply (mm) |
| sw_demand | Demand for soil water (mm) |
| sw_deficit | Soil water deficit (mm) |
| vpd_est | Estimated vapor pressure deficit (kPa) |
| esw_layr | Extractable soil water (mm) |
| dlt_dm | Actual above-ground dry matter production ($g/m^2$) |
| dlt_dm_pot_rue | Potential above-ground dry matter production via photosynthesis ($g/m^2$) |

### 2.4. Data Preprocessing

Before training machine learning predictive models using the dataset, it is crucial to ensure the data's effectiveness for machine learning model training [42,43]. This involves some preprocessing steps on the dataset and the output data from the APSIM. The preprocessing steps include aggregating data from all sources to construct a new dataset as input for the ML model. Subsequently, this new dataset undergoes unit conversion, handling missing values, the removal of anomalies, and addressing data gaps. For features lacking data, new feature variables might be constructed to enhance interpretability and overall model performance. Following this, normalization processes are applied to ensure consistent scales and ranges across features. Finally, feature selection is performed based

on their importance, selecting multiple crucial features as the ultimate input for the ML predictive model.

### 2.4.1. Aggregation of All Data Sources and Unit Conversions

The dataset comprises 35 variables, including output variables from the APSIM model, wheat crop yield data collected over 39 years from both historical yearbooks of the Gansu Provincial Bureau of Statistics (gansu.gov.cn, accessed on 10 March 2023) and actual measurements, meteorological data from 1984 to 2021 obtained from the Gansu Provincial Meteorological Bureau (cma.gov.cn, accessed on 10 March 2023), soil property data from experimental fields, and on-field crop management practices. All data units for wheat seeding quantity, actual yield, fertilization amount, etc., were standardized to (kg/ha). The units of thirteen soil characteristics at nine different depths of the soil were converted to values under soil depth in millimeters. These variables, encompassing the target variable, served as the input data for both the ML prediction model and model testing. Factors influencing crop growth include changes in growth environment, human management, and crop genotype variation [44]. Changes in growth environment and human management have specific, detailed parameter data, whereas crop genotype variation is influenced by multiple factors, including advancements in biological and cultivation techniques over the years and the genetic evolution of crops due to environmental adaptation, leading to an increasing yield trend. To address the absence of publicly available genotype datasets, this experiment constructed new feature variables to interpret the changes in crop genotype, elaborated in detail in "Section 2.4.3".

### 2.4.2. Imputation of Missing Values and Removal of Outliers

Handling outliers and filling missing values are crucial steps in preparing data for machine learning modeling [45]. There are several methods to address missing values [46]. In this experiment, missing data refer to soil data and planting management data that were not recorded before 2015. Based on the soil characteristic values recorded from 2015 to 2021 at six depth levels ranging from 500 mm to 2000 mm, which remained unchanged, we used the mode of the measurements to interpolate the missing soil data. For soil characteristics at depths ranging from 0 mm to 300 mm, we interpolated the data using the average values recorded from 2015 to 2021. As the planting in the experimental fields was uniformly managed, missing planting management data were completed based on standard management practices. Outliers refer to data points that deviate significantly from normal values due to measurement instrument errors or recording mistakes during soil measurement. These outliers were deleted, and the mode of the measurements from the same group was used to fill in the gaps.

### 2.4.3. Construction of Features

By observing the historical wheat production data and fitting a curve against the years, we clearly see a consistent upward trend in wheat production over time, as depicted in Figure 3, and Figure 4 illustrates the regression residuals of wheat production over time. To quantitatively describe the trend of wheat yield over time, we introduced a new feature trend value called "Annual Growth Rate" $Year_{i(year)}$, representing the positive impact of genetic improvement on yield. Subsequently, we introduced another new feature trend value, called "Annual Trend Prediction" $\hat{y}$, to describe the changing trend of wheat yield over time. It is worth emphasizing that constructing new features only involves datasets that do not include the test year. Our goal was to use the model to predict the data in the test set. Specifically, we set the trend value for each year in the test set as the model's predicted yield for that year, which can be represented by the following formula:

$$\hat{y} = b_0 Year_{i(year)} + b_1 \tag{1}$$

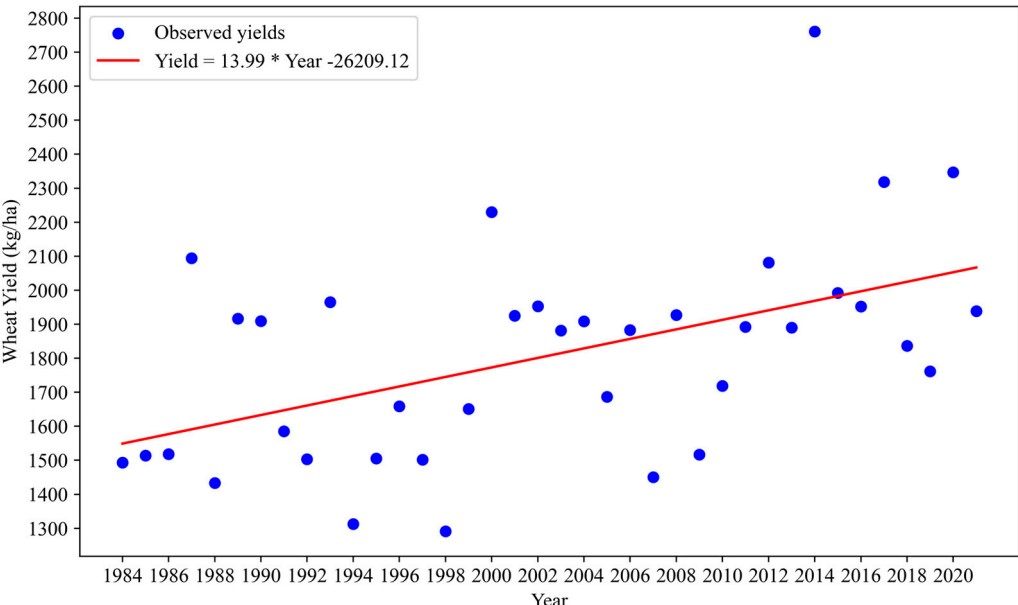

**Figure 3.** Linear-fitting regression plot of time and the observed yield of wheat. Note: This graph only depicts the stable increasing trend of wheat yield over time, and the fitted curve in the graph is not related to genetic variables.

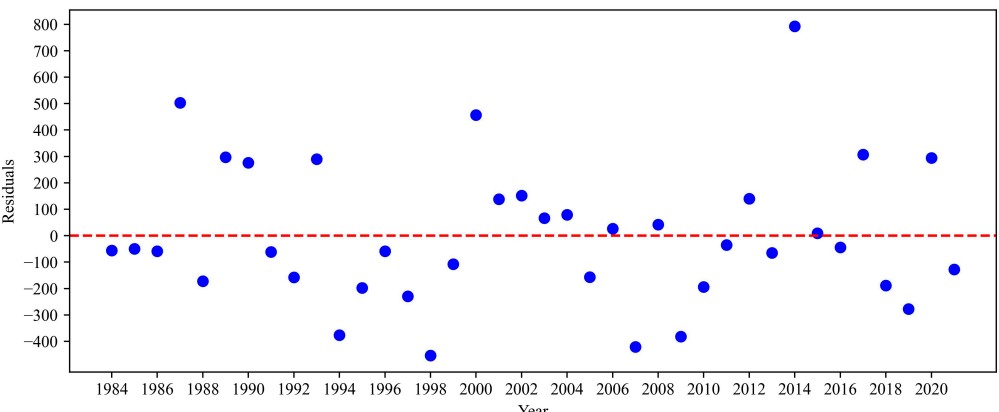

**Figure 4.** Map of the wheat yield over time and regression surrounding residuals. The red dashed line represents the horizontal reference line in the residual plot, and the blue dots represents the residual for each sample, which is the difference between the true value and the model prediction.

Through this method, we successfully introduced predicted production values as new features into the model to better explain the changing trend of wheat production over time.

### 2.4.4. Normalization of the Feature Data

Some feature data in the dataset exhibit relatively large numerical values, which, when used to train predictive ML models, particularly weighted ensemble models, may cause the models to exhibit a bias towards these larger-value features [43]. To address the potential issue of the model focusing more on features with larger values and less on other features, we scaled all values of the input data for the features 'carbon_tot', 'biomass', and 'nit_tot' using the min–max scaling method to normalize them between 0 and 1. This ensures that all features have similar value ranges while preserving the distribution characteristics of the input variables [47]. Min–max scaling is implemented using the following formula:

$$X_{norm} = (X - X_{min}) / X_{max} - X_{min} \tag{2}$$

　　　　Scaling all feature values to a similar range ensures a more balanced impact of each feature on the model.

### 2.4.5. Feature Selection

　　　　The machine learning predictive model comprises a vast array of input features, totaling up to 1206, including APSIM output variables, weather data, soil information, engineered features, and management practices. Due to the dataset's enormity and the risk of overfitting with high-dimensional training data, we only applied feature selection measures to the training dataset to ensure the performance and generalization ability of the machine learning model. Initially, we relied on domain experts' knowledge to conduct feature selection based on their professional insights. Subsequently, we employed permutation feature selection methods to further refine the features.

### Expert Knowledge-Based Feature Selection

　　　　We first evaluated the correlation between each feature outputted by the APSIM and wheat yield. Then, combining the domain experts' advice, we excluded features that, although statistically correlated, were unlikely to practically the impact yield. We excluded features between the end of harvesting and the beginning of planting for the next year to reduce the number of weather-related features. Additionally, we excluded cumulative planting progress data from the first few weeks before planting as they did not provide substantive information for the model. Furthermore, we paid special attention to the temporal features of non-zero yield variables, as they directly relate to critical stages of the crop growth cycle. Therefore, utilizing professional knowledge, we ultimately retained 100 characteristic variables, encompassing various dimensions including biomass accumulation, leaf area development, soil moisture supply, nitrogen absorption, etc. These variables reflect wheat's response to environmental conditions, changes in physiological state, and sensitivity to management measures. Ultimately, based on expert knowledge, the feature selection process reduced the number of features from 1206 to 100 (Table 5).

**Table 5.** The 100 features selected based on expert knowledge.

| | | | | | | | | | | |
|---|---|---|---|---|---|---|---|---|---|---|
| day_of_year | grain_n_demand | grain_size | radn | sws | trend | grain_size | GrowthN | leaf_no | n_demand_head | |
| yield | sw_stress_photo | grain_wt | maxt | dul | dnit | grain_protein | GrowthP | LeafDetachingN | n_supply_soil | |
| biomass | sw_stress_expan | Grain1GrainTotalN | mint | sat | cover_green | Grain1DetachingN | GrowthWt | LeafDetachingP | n_uptake | |
| root_depth | water_table | sw_supply | rain | swcon | dlt_lai | Grain1DetachingP | HeadGreenWt | LeafDetachingWt | p_conc_stover | |
| flowering_date | runoff | sw_demand | day_length | nit_tot | dlt_n_fixed | Grain1DetachingWt | HeadGrowthN | LeafTotalP | p_demand | |
| maturity_date | dlt_n_min_tot | sw_deficit | vp | ll15_dep | dlt_Pai | green_biomass | HeadGrowthP | LeafTotalpconc | p_uptake | |
| lai | nitrification | vpd_est | air_dry | ll_dep | dm_plant_min | green_biomass_n | HeadGrowthWt | n_conc_crit | Pai | |
| ep | dlt_n_min | esw_layr | bd | LL15 | effective_rue | green_biomass_p | lai_sum | n_conc_crit_grain | SenescedN | |
| es | grain_n | dlt_dm | carbon_tot | PH | grain_p | green_biomass_wt | leaf_area | n_conc_grain | sowing_date | |
| n_demanded | grain_protein | dlt_dm_pot_rue | esw | fertiliser | grain_wt | GreenWt | TTAfteremergence | n_demand_grain | Stage | |

### Permutation Feature Selection and Random Forests

　　　　When dealing with the input features of varying measurement scales or category counts, the default variable importance in random forests based on impurity might show bias [39]. Permutation feature importance assessment involves shuffling individual features and observing changes in model performance to gauge their contribution to the model's predictions [48]. Combining permutation-based feature selection with random forest methods, this approach accurately evaluates feature importance by shuffling feature values in the test set for each feature. This method ensures a more stable determination of feature importance sequence, particularly in complex data [49–51]. This comprehensive approach offers a more accurate assessment of feature importance, especially demonstrating better stability and consistency when handling complex data.

　　　　In our experiment combining permutation feature selection and random forests, we trained a random forest model with 100 sub-trees. Through tenfold cross-validation, we adjusted the number of trees and ultimately selected the top 28 ranked features as inputs for the machine learning model. This effectively filtered out a set of features signifi-cantly impacting the model's performance, laying a reliable foundation for subsequent predictive modeling. This method enabled us to precisely identify which features con-

tribute most to the model's predictions, thereby enhancing the model's performance and generalization ability.

### 2.5. Adjustment of Hyperparameters

The appropriate hyperparameters can significantly enhance a model's performance, and different combinations might yield varying results for the same data. In the process of constructing high-performance machine learning models using training data, we adopted a strategy that combined 20 iterations of Bayesian search with 10-fold cross-validation to optimize hyperparameter tuning [52]. Bayesian search was used to guide the search for hyperparameters, while cross-validation was used to evaluate the performance of hyperparameters. This combination method effectively navigated the hyperparameter space to determine the optimal configuration [53–55].

### 2.6. Predictive Models

#### 2.6.1. Agricultural Production Simulator (APSIM)

The APSIM (Agricultural Production Systems sIMulator) is a globally renowned agricultural production system simulator developed by the Commonwealth Scientific and Industrial Research Organisation (CSIRO) of Australia. It is designed for assessing and optimizing the production and environmental outcomes of agricultural systems [56]. The APSIM simulator utilizes mathematical models and physical descriptions of agricultural production systems to simulate the behavior of agroecosystems. This includes simulating processes such as field management, soil carbon–nitrogen balance, soil water balance, crop yield, phenology, and photosynthesis [57,58]. In this study, we employed the Wheat module from the APSIM version 7.10 for experimental sites in the southern–central region of Gansu, China. Specifically, the Wheat module of APSIM is dedicated to simulate wheat growth and yield formation [59]. Additionally, we utilized the Soil module to simulate dynamic changes in soil moisture and nutrients [60], the Weather module for providing meteorological data as input for simulations [12], and the Management module to describe and manage agricultural practices and measures [61]. These modules operate synchronously on a daily time step.

#### 2.6.2. Eight Machine Learning Algorithms

To ensure the construction of a high-performance ensemble model, we utilized the APSIM alongside a range of diverse and reliable base learners, including multiple linear regression [62], LASSO regression [63,64], decision tree regression (DTR) [65], random forest (RF) [49,66,67], XGBoost [68], LightGBM [69], gradient boosting regression (GBR) [70,71], and extremely randomized trees regression (ERT) [72,73]. We optimized these eight machine learning algorithms through weighted ensemble techniques [74]. Emphasizing the diversity among these base learners and their individual strong performances, we intelligently combined them after thorough training into multiple high-performance and robust machine learning models [74]. Ultimately, by comparing the performance of single models to the ensemble model, we selected the ensemble model that exhibited the best predictive performances.

#### 2.6.3. Weighted Average Ensemble Model

The weighted average ensemble method, commonly used to enhance the model performance, was employed in this study. It involves combining the predictions from the eight base learners by weighted averaging to obtain the final prediction. Each base learner carries an equal weight, resulting in a final prediction that corresponds to the mean of all model predictions [75]. However, to ensure the performance of the ensemble model, it is typically crucial to select base learners with high diversity, ensuring differences among the models [76,77].

2.6.4. Optimized Weighted Ensemble Model

The optimized weighted ensemble model, as opposed to a simple weighted average ensemble, considers the performance of base models on the training set to dynamically adjust the allocation of model weights. This approach allows higher-performance learners to have greater weights, thereby enhancing the overall model accuracy [74,78]. Studies indicate that the optimized weighted ensemble model outperforms the simple weighted average model in terms of model performance. This optimization is achieved by minimizing the mean squared error (MSE), root-mean-square error (RMSE), and the coefficient of determination (R-squared). Based on the comparison and analysis of the experimental results with three different objective functions ("Section 4"), the weighted ensemble model optimized by the MSE for predictions and superior model performance was selected. In large-scale datasets, optimizing weights based on the MSE aids in adjusting the model complexity by optimizing the square of the error, which helps to prevent overfitting to some extent, thereby enhancing the model robustness and performance. This optimization problem is addressed by adjusting model parameters using gradient descent algorithms to minimize the MSE, which is equivalent to minimizing the loss function. The following is the fundamental concept of gradient descent:

i.    Set the model parameters (weights) to certain values.
ii.   Compute the gradient of the loss function with respect to the parameters.
iii.  Update the parameters in the direction opposite to the gradient to reduce the loss.
iv.   Repeat the above steps until the loss reaches a satisfactory level or the maximum number of iterations is reached.

The mathematical representation of gradient descent is as follows:

$$\hat{y}_j = \sum_{i=1}^{n} W_i \hat{y}_{ij} \tag{3}$$

$$\sum_{i=1}^{n} W_i = 1 \tag{4}$$

$$W_i = W_i - \alpha \frac{\partial}{\partial W_i} MSE \tag{5}$$

$$\frac{\partial}{\partial W_i} MSE = \frac{1}{N} \sum_{i=1}^{N} (y_j - \hat{y}_j)^2 \tag{6}$$

$$W_i \geq 0, \ \forall i = 1, \ldots, 8 \tag{7}$$

where $W_i$ is the weight of the $i$ model, $\hat{y}_{ij}$ is the prediction of the $i$ model for the $j$ sample in the entire dataset ($j = 1, \ldots, k$), $\alpha$ is the learning rate controlling the update step size, $MSE$ is the mean squared error, $\frac{\partial}{\partial W_i} MSE$ is the partial derivative with respect to weight $W_i$, $y_j$ is the actual observed value, $\hat{y}_j$ is the prediction of the entire model, which is the weighted sum of all model predictions. The optimal weights are determined by solving the minimization problem of the entire model (Equations (3)–(6)) while satisfying the constraint that the sum of all weights must be one (Equation (4)).

*2.7. Evaluation Metrics*

To assess the performance of the developed machine learning models, we utilized four statistical evaluation metrics: the root-mean-square error (*RMSE*), the relative root-mean-square error (*RRMSE*), the mean bias error (*MBE*), and the coefficient of determination (*R-squared*) (Equations (8)–(11), respectively). These metrics represent the accuracy, robustness, and explanatory power of the model predictions [79].

The root-mean-square error (*RMSE*) quantifies the disparity between the predicted and actual values, where a lower value signifies a more accurate model fit:

$$RMSE = \sqrt{\frac{1}{n} \sum_{i=1}^{n} (y_i - \hat{y}_i)^2} \tag{8}$$

The relative root-mean-square error (*RRMSE*) quantifies the model's prediction error relative to the variation in the target variable; a smaller *RRMSE* signifies a smaller prediction error concerning the variability of the target variable, indicating a better predictive performance:

$$RRMSE = \frac{RMSE}{\frac{1}{n}\sum_{i=1}^{n} y_i} \tag{9}$$

The mean bias error (*MBE*) describes the average deviation in predictions. A negative MBE implies the model underestimates the target variable, a positive value indicates overestimation, and values close to zero signify smaller average deviations between the predictions and observations:

$$MBE = \frac{1}{n}\sum_{i=1}^{n}(\hat{y}_i - y_i) \tag{10}$$

The coefficient of determination (*R-squared*) represents the model's ability to explain the variability in observed values. It ranges from 0 to 1, where values closer to 1 denote a better model fit, while values near 0 signify a poorer model performance:

$$R^2 = 1 - \frac{RSS(sun\ of\ square\ residuals)}{TSS(Total\ sum\ of\ sqaures)} = 1 - \frac{\sum_{i=1}^{n}(y_i - \hat{y}_i)^2}{\sum_{i=1}^{n}(y_i - \overline{y})^2} \tag{11}$$

After computing and comparing these four statistical evaluation metrics, we identified the model with the best overall performance. We conducted a detailed analysis of its results to ensure interpretability [79].

### 2.8. Experimental Environment Configuration

The research was conducted using a Windows 11 platform with a Python 3.8.5 environment, employing a suite of Python libraries for data processing, analysis, and machine learning modeling. The hardware specifications included an Intel Core i7-9700K CPU @ 3.60 GHz, 16 GB RAM, and an NVIDIA GeForce RTX 2080 Ti GPU. Data processing and analysis were facilitated by the pandas library, with sklearn.preprocessing.MinMaxScaler utilized for data scaling and numpy for numerical computations, while the os module was employed for operating system interactions. In terms of machine learning modeling, various models from Scikit-learn were utilized, including ensemble learning models (RandomForestRegressor, GradientBoostingRegressor, ExtraTreesRegressor), linear models (Lasso, LinearRegression), decision tree models (DecisionTreeRegressor), as well as regression models from gradient boosting frameworks (XGBRegressor, LGBMRegressor). Model performance evaluation relied on Scikit-learn's mean_squared_error and r$^2$_score functions. Graphical visualization was conducted using Microsoft Word 2021 and Matplotlib 3.3.5. To ensure experiment reproducibility and stability, Python virtual environments were employed to isolate dependencies for different projects, with package versions managed via the pip tool.

## 3. Results

### 3.1. APSIM Model Performance Verification

Based on field experiment data from 1984 to 2021, we utilized the APSIM model to predict wheat yields for 38 years. Analyzing the relationship between the predicted and actual yields (Figure 5), the performance metrics for the APSIM model were as follows: RMSE: 169.82 (kg/ha), RRMSE: 9.39%, MBE: −4.96 (kg/ha), and R-squared: 0.70. With the RRMSE of the APSIM model within 10%, the process of creating daily wheat yield data based on the APSIM simulation from the actual yearly wheat yields is scientifically robust.

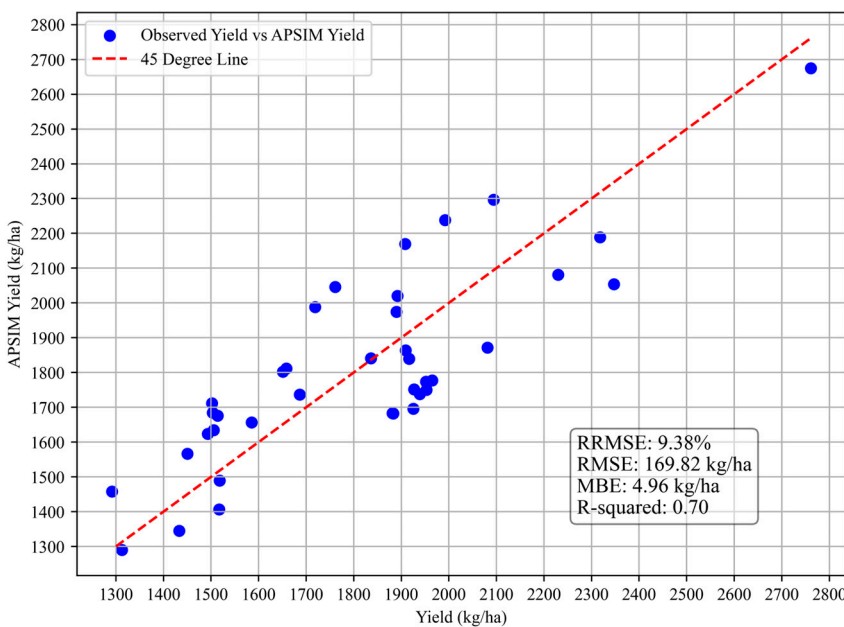

**Figure 5.** The relationship between the wheat yield predicted by the APSIM model and the actual observed wheat yield from 1984 to 2021. The horizontal coordinate of the blue dots represents the actual wheat yield, and the vertical coordinate represents the yield predicted by the APSIM model. The red 45-degree dashed line represents the ideal situation where the predicted values are completely consistent with the actual values.

### 3.2. Prediction of the Model Performance

This study employed an extensive dataset covering wheat production spanning from 1984 to 2021, comprising data from 38 years. Specifically, for a comprehensive assessment, we used Python's random.sample() function to randomly select 3 different years from among 38 years, which resulted in 2012, 2016, and 2021, acting as the test dataset for assessing the effectiveness of various models. Through an evaluation of the performance of a single APSIM prediction model, twelve machine learning prediction models, and twelve APSIM-based machine learning prediction models, we obtained a range of error metrics and explained variances for the experimental results.

Table 6 demonstrates the test set prediction errors of the APSIM prediction model across the three different years. Specifically, for the 2021 test set, the RRMSE and MBE were 10.69% and 107.41 (kg/ha), respectively. For the 2016 test set, the RRMSE and MBE were 9.93% and 106.30 (kg/ha), respectively. Lastly, for the 2012 test set, the RRMSE and MBE were 11.71% and 108.71 (kg/ha), respectively.

**Table 6.** APSIM model's test evaluation standard error values for the years of 2012, 2016, and 2021.

| Year | RMSE (kg/ha) | RRMSE (%) | MBE (kg/ha) | $R^2$ |
|------|--------------|-----------|-------------|-------|
| 2012 | 134.11 | 10.69 | 107.41 | 0.9573 |
| 2016 | 133.63 | 9.93 | 106.30 | 0.9616 |
| 2021 | 135.67 | 11.71 | 108.71 | 0.9613 |

In the test results for 2012 shown in Table 7, for the ML predictive models, the RMSE ranged from 130.06 to 261.94 (kg/ha), RRMSE ranged from 10.33% to 20.92%, the MBE ranged from 33.58 to 242.10 (kg/ha), and the $R^2$ ranged from 0.83 to 0.96. As for the APSIM-ML predictive models, the RMSE ranged from 26.87 to 146.18 (kg/ha), the RRMSE ranged from 2.12% to 11.69%, the MBE ranged from −0.53 to 116.65 (kg/ha), and the $R^2$ ranged from 0.94 to 0.99. The 'Increase in RRMSE (%)' column in Tables 7–9 represents the change in numerical values.

**Table 7.** Test evaluation errors for the benchmark ML and APSIM-ML models in 2012.

| ML Model | RMSE (kg/ha) | RRMSE (%) | MBE (kg/ha) | $R^2$ | APSIM-ML Model | RMSE (kg/ha) | RRMSE (%) | MBE (kg/ha) | $R^2$ | Increase in RRMSE (%) |
|---|---|---|---|---|---|---|---|---|---|---|
| Random Forest | 231.68 | 18.52 | 199.71 | 0.8744 | APSIM-Random Forest | 111.62 | 8.91 | 94.31 | 0.9737 | −9.61 |
| Lasso Regression | 208.63 | 16.72 | 107.07 | 0.8952 | APSIM-Lasso Regression | 146.09 | 11.69 | 116.65 | 0.9462 | −5.03 |
| Decision Tree | 260.54 | 20.92 | 208.15 | 0.8376 | APSIM-Decision Tree | 144.71 | 11.58 | 113.63 | 0.9463 | −9.34 |
| Linear Regression | 259.41 | 20.74 | 242.1 | 0.8375 | APSIM-Linear Regression | 97.92 | 7.83 | 79.83 | 0.9742 | −12.91 |
| Gradient Boosting | 234.04 | 18.76 | 200.57 | 0.8702 | APSIM-Gradient Boosting | 94.62 | 7.59 | 74.22 | 0.9752 | −11.17 |
| Extra Trees | 228.07 | 18.07 | 186.7 | 0.8767 | APSIM-Extra Trees | 130.89 | 10.47 | 110.43 | 0.9608 | −7.60 |
| XGBoost | 229.76 | 18.36 | 196.47 | 0.8743 | APSIM-XGBoost | 99.80 | 7.96 | 78.26 | 0.9760 | −10.4 |
| LightGBM | 229.18 | 18.24 | 191.82 | 0.8751 | APSIM-LightGBM | 105.05 | 8.39 | 78.68 | 0.9743 | −9.85 |
| Weighted Average Ensemble Minimized MSE | 225.75 | 17.98 | 191.83 | 0.8785 | Weighted Average Ensemble Minimized MSE | 110.95 | 8.87 | 93.19 | 0.9713 | −9.11 |
| Optimized Weighted Integrations Minimized RMSE | 129.53 | 10.33 | 33.58 | 0.9608 | Optimized Weighted Integrations Minimized RMSE | 26.78 | 2.12 | −0.53 | 0.9978 | −8.21 |
| Optimized Weighted Integrations Minimized $R^2$ | 133.24 | 10.56 | 36.74 | 0.9577 | Optimized Weighted Integrations Minimized $R^2$ | 47.66 | 3.56 | 4.25 | 0.9948 | −7.00 |
| Optimized Weighted Integrations | 139.41 | 11.09 | 34.27 | 0.9541 | Optimized Weighted Integrations | 47.93 | 3.55 | 5.14 | 0.9947 | −7.54 |

**Table 8.** Test evaluation errors for the benchmark ML and APSIM-ML models in 2016.

| ML Model | RMSE (kg/ha) | RRMSE (%) | MBE (kg/ha) | $R^2$ | APSIM-ML Model | RMSE (kg/ha) | RRMSE (%) | MBE (kg/ha) | $R^2$ | Increase in RRMSE (%) |
|---|---|---|---|---|---|---|---|---|---|---|
| Random Forest | 142.76 | 10.63 | 115.76 | 0.9347 | APSIM-Random Forest | 79.15 | 5.89 | 58.93 | 0.9827 | −4.74 |
| Lasso Regression | 296.16 | 21.79 | 154.49 | 0.8045 | APSIM-Lasso Regression | 150.68 | 11.11 | 120.35 | 0.952 | −10.68 |
| Decision Tree | 217.07 | 15.5 | 90.13 | 0.9013 | APSIM-Decision Tree | 147.05 | 10.91 | −5.67 | 0.9549 | −4.59 |
| Linear Regression | 83.93 | 6.57 | 54.02 | 0.9821 | APSIM-Linear Regression | 92.07 | 6.88 | 67.51 | 0.9733 | 0.31 |
| Gradient Boosting | 123.88 | 9.65 | 90.49 | 0.9656 | APSIM-Gradient Boosting | 66.69 | 4.94 | 24.95 | 0.9806 | −4.71 |
| Extra Trees | 173.15 | 12.68 | 144.89 | 0.9405 | APSIM-Extra Trees | 98.92 | 7.28 | 73.77 | 0.96 | −5.4 |
| XGBoost | 128.8 | 9.13 | 91.04 | 0.9657 | APSIM-XGBoost | 67.36 | 5.05 | 25.87 | 0.9805 | −4.08 |
| LightGBM | 108.21 | 8.17 | 96.9 | 0.9744 | APSIM-LightGBM | 67.63 | 5.05 | 21.01 | 0.9804 | −3.12 |
| Weighted Average Ensemble Minimized MSE | 118.61 | 9.92 | −39.3 | 0.9725 | Weighted Average Ensemble Minimized MSE | 77.09 | 5.72 | 57.88 | 0.9774 | −4.2 |
| Optimized Weighted Integrations Minimized RMSE | 44.75 | 3.47 | 10.05 | 0.9952 | Optimized Weighted Integrations Minimized RMSE | 50.58 | 3.83 | 6.71 | 0.9916 | 0.36 |
| Optimized Weighted Integrations Minimized $R^2$ | 45.19 | 3.47 | 10.00 | 0.9952 | Optimized Weighted Integrations Minimized $R^2$ | 47.22 | 3.52 | 4.37 | 0.9925 | 0.05 |
| Optimized Weighted Integrations | 45.02 | 3.41 | 10.33 | 0.9952 | Optimized Weighted Integrations | 47.44 | 3.56 | 5.09 | 0.9924 | 0.15 |

**Table 9.** Test evaluation errors for the benchmark ML and APSIM-ML models in 2021.

| ML Model | RMSE (kg/ha) | RRMSE (%) | MBE (kg/ha) | $R^2$ | APSIM-ML Model | RMSE (kg/ha) | RRMSE (%) | MBE (kg/ha) | $R^2$ | Increase in RRMSE (%) |
|---|---|---|---|---|---|---|---|---|---|---|
| Random Forest | 117.56 | 10.78 | −49.81 | 0.9718 | APSIM-Random Forest | 123.73 | 10.75 | 97.20 | 0.9684 | −0.03 |
| Lasso Regression | 166.21 | 15.35 | −41.83 | 0.9314 | APSIM-Lasso Regression | 106.89 | 9.21 | 78.55 | 0.9756 | −6.14 |
| Decision Tree | 209.03 | 19.07 | −21.97 | 0.8971 | APSIM-Decision Tree | 160.26 | 13.87 | 49.86 | 0.9453 | −5.2 |
| Linear Regression | 93.09 | 8.51 | −70.52 | 0.9846 | APSIM-Linear Regression | 98.07 | 8.44 | 57.45 | 0.9792 | −0.07 |
| Gradient Boosting | 114.01 | 10.39 | −25.4 | 0.9744 | APSIM-Gradient Boosting | 67.78 | 5.83 | 17.96 | 0.9902 | −4.56 |
| Extra Trees | 119.38 | 10.92 | −28.05 | 0.9661 | APSIM-Extra Trees | 146.34 | 12.64 | 116.25 | 0.9559 | 1.72 |
| XGBoost | 121.62 | 11.11 | −44.84 | 0.9643 | APSIM-XGBoost | 60.62 | 5.25 | 21.45 | 0.9921 | −5.86 |
| LightGBM | 85.56 | 7.87 | −34.98 | 0.9822 | APSIM-LightGBM | 58.36 | 5.1 | 17.28 | 0.9921 | −2.77 |
| Weighted Average Ensemble | 107.90 | 9.89 | −39.27 | 0.9701 | Weighted Average Ensemble | 86.56 | 7.46 | 56.88 | 0.9833 | −2.43 |
| Minimized MSE optimized weighted integrations | 70.07 | 6.53 | −11.71 | 0.9883 | Minimized MSE Optimized Weighted Integrations | 43.3 | 3.74 | −2.53 | 0.9957 | −2.79 |
| Minimized RMSE Optimized Weighted Integrations | 75.61 | 7.07 | −10.10 | 0.9852 | Minimized RMSE Optimized Weighted Integrations | 47.59 | 3.54 | 4.24 | 0.9953 | −3.53 |
| Minimized $R^2$ Optimized Weighted Integrations | 70.94 | 6.53 | −8.34 | 0.9869 | Minimized $R^2$ Optimized Weighted Integrations | 47.88 | 3.56 | 4.95 | 0.9950 | −2.97 |

The test results for 2016 displayed in Table 8 indicate that, for the ML predictive models, the RMSE ranges from 44.75 to 296.16 (kg/ha), the RRMSE ranges from 3.47% to 21.79%, the MBE ranges from −39.30 to 154.49 (kg/ha), and the $R^2$ ranges from 0.80 to 0.99. Concerning the APSIM-ML predictive models, the RMSE ranges from 47.22 to 150.68 (kg/ha), the RRMSE ranges from 3.52% to 11.11%, the MBE ranges from −5.67 to 120.35 (kg/ha), and the $R^2$ ranges from 0.95 to 0.99.

In the 2021 test results shown in Table 9, for the ML prediction models, the RMSE ranges from 70.07 to 209.03 (kg/ha), the RRMSE spans from 6.53% to 19.07%, the MBE varies from −8.34 to −70.52 (kg/ha), and the $R^2$ ranges from 0.89 to 0.98. For the APSIM-ML prediction models, the RMSE varies from 43.30 to 160.26 (kg/ha), the RRMSE spans from 3.54% to 13.87%, the MBE ranges from −2.50 to 116.25 (kg/ha), and the $R^2$ varies from 0.94 to 0.99.

### 3.3. Comparison and Analysis of Model Performance in Different Years

By analyzing Table 7 and Figure 6 for the 2012 prediction data, we observe that combining APSIM output data as input features for the APSIM-ML predictive model significantly improves the yield prediction performance compared to the 12 developed ML models and the standalone APSIM model. The APSIM-ML model exhibited an average decrease in the RRMSE by −8.98% and an average reduction in the RMSE by −111.97 (kg/ha) compared to the ML models. Among these, the MLR predictive model demonstrated the highest improvement in RRMSE precision by 12.91%. Post-MSE optimization, the APSIM-ML predictive model displayed superior predictive performance with an 8.84% RRMSE enhancement compared to the APSIM predictive model and an 18.80% and 10.33% RRMSE enhancement against the worst and best performing ML predictive models, respectively. In the experimental results, individual ML predictive models exhibit the poorest wheat yield prediction performance. However, the weighted ensemble of multiple ML models demonstrates a higher predictive accuracy compared to the individual ML models, resulting in RRMSE reductions ranging from 9.02% to 10.59%.

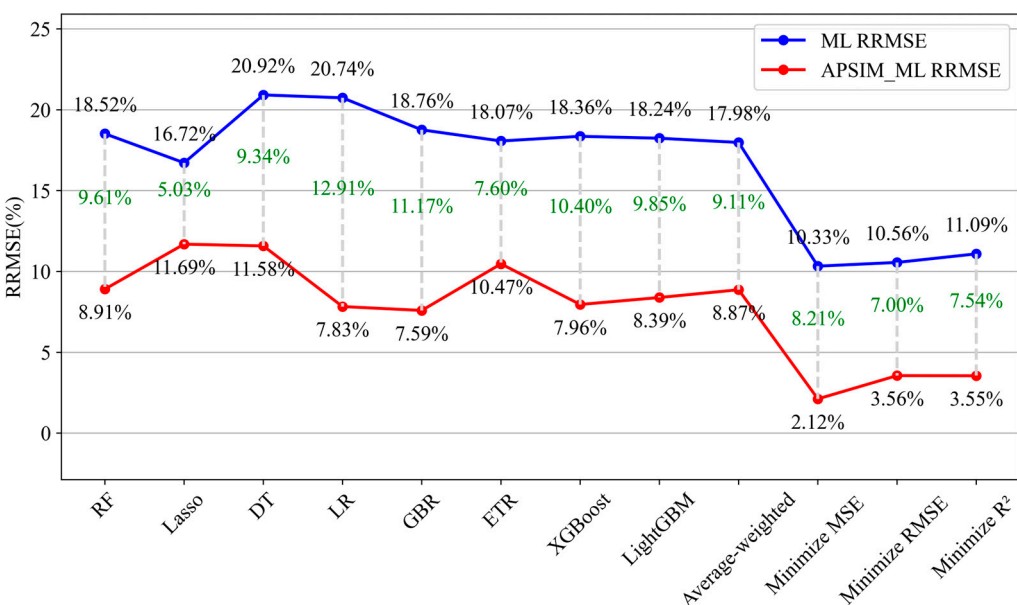

**Figure 6.** Comparison of the RRMSE test results between the 2012 ML predictive models and the APSIM-ML predictive model.

Moreover, the optimized weighted ensemble model outperformed the weighted average ensemble predictive model. Regarding the predictive performance of the ML models, the ranking from the lowest to the highest was DT < MLR < GBR < RF < XGBoost < LightGBM < ETR < Weighted average < Lasso < Minimized $R^2$ < Minimized RMSE < Minimized MSE. In contrast, for the APSIM-ML models, the predictive performance ranking was Lasso

< DT < ETR < RF < Weighted average < LightGBM < XGBoost < MLR < GBR < Minimized RMSE < Minimized $R^2$ < Minimized MSE.

By analyzing the RRMSE test results for 2016 (Table 8 and Figure 7), it is evident that incorporating the APSIM variables improved all constructed ML predictive models, except for MLR. In comparison to the ML models, the APSIM-ML predictive model demonstrated an average RRMSE enhancement of −3.38% and an average RMSE improvement of −44.64 (kg/ha). However, the RRMSE for the MLR predictive model decreased by 0.31% when compared to APSIM-MLR. The APSIM-ML predictive model with the optimized weighted ensemble model, minimizing the RMSE, displayed the best performance (RRMSE = 3.52%) among the models in 2016. The RRMSE for the worst and best ML predictive models compared to APSIM-ML increased by −5.50%, −18.27%, and 0.11%, respectively. Notably, the ML optimized weighted ensemble predictive model outperformed the APSIM-ML optimized ensemble predictive model in the 2016 dataset. In terms of the predictive performance for the ML models: Lasso < DT < ETR < RF < Weighted average < GBR < XGBoost < LightGBM < MLR < Minimized RMSE < Minimized MSE < Minimized $R^2$; and for the APSIM-ML models: Lasso < DT < ETR < MLR < RF < Weighted average < XGBoost < LightGBM < GBR < Minimized MSE < Minimized $R^2$ < Minimized RMSE.

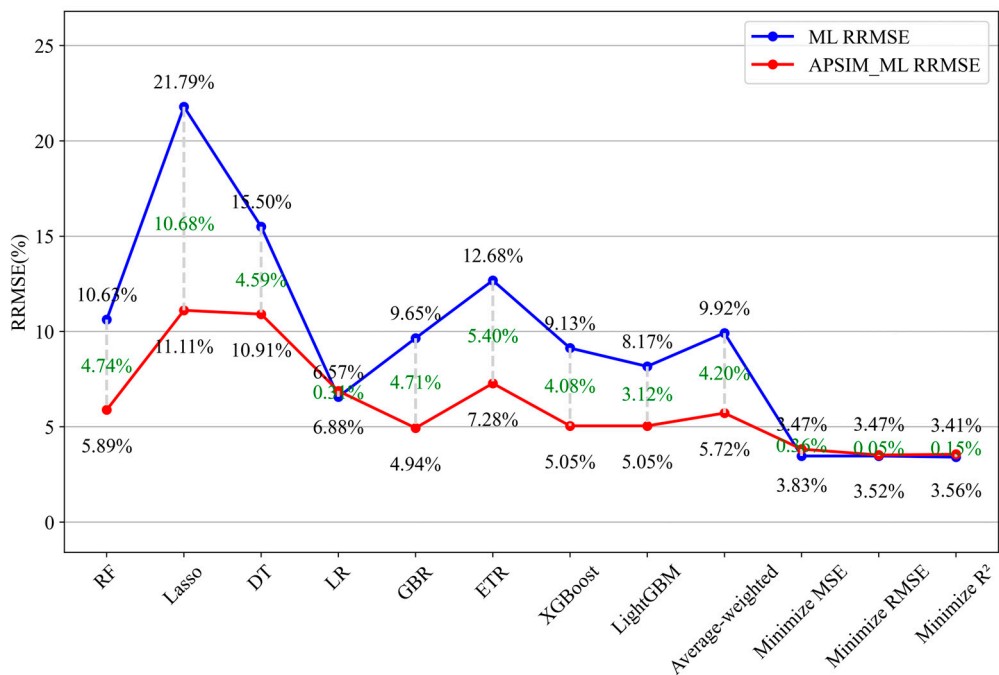

**Figure 7.** Comparison of the RRMSE test results between the 2016 ML predictive models and the APSIM-ML predictive model.

By analyzing the predictive outcomes for the various models in 2021 (Table 9 and Figure 8), we observed that the APSIM-ML model exhibited an average RRMSE and RMSE improvement of −2.88% and −25.30 (kg/ha), respectively, compared to the ML models. However, the RRMSE for the ETR model increased by −1.72% when compared to APSIM-ETR. The APSIM-ML predictive model with the optimized weighted ensemble, minimizing RMSE, displayed the best performance (RRMSE = 3.54%) when compared to the APSIM model. The worst and best ML predictive models had an RRMSE increase of −8.20%, −15.53%, and −2.99%, respectively. For the ML models' predictive performance ranking: DT < Lasso < XGBoost < ETR < RF < GBR < Weighted average < MLR < LightGBM < Minimized RMSE < Minimized $R^2$ < Minimized MSE; and for the APSIM-ML models' predictive performance: DT < ETR < RF < Lasso < MLR < Weighted average < GBR < XGBoost < LightGBM < Minimized MSE < Minimized $R^2$ < Minimized RMSE.

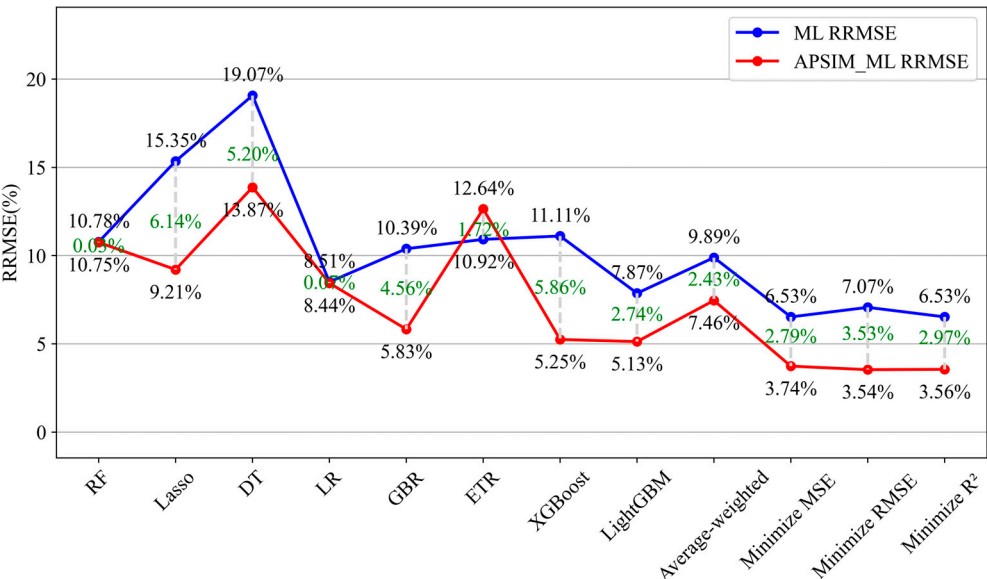

**Figure 8.** Comparison of the RRMSE test results between the 2021 ML predictive models and the APSIM-ML predictive model.

*3.4. Relationship between APSIM-ML Model Performance and Input Features*

When analyzing the relationship between the variables and the RRMSE, it was observed that the eight APSIM-ML predictive models exhibited similar relationships for each test year. The average correlation heatmap and retrograde analysis of the three correlation matrices between certain APSIM output variables, constructed genotype variables (trend), weather features, soil variables, and the RRMSE of the various machine learning predictive models are presented in Figures 9 and 10. In the heatmap, each cell contains a numerical value representing the correlation coefficient between two variables. The correlation coefficient ranges for the analysis are categorized as follows: −0.20 to 0.20 indicates a weak correlation, 0.20 to 0.50 suggests a moderately positive correlation, 0.50 to 0.80 indicates a strong positive correlation, −0.20 to −0.50 denotes a moderately negative correlation, −0.50 to −0.80 indicates a strong negative correlation, 0.8 to 1 signifies a highly positive correlation, −0.8 to −1 represents a highly negative correlation, 1 indicates a perfect positive correlation, 0 implies no linear correlation, and −1 signifies a perfect negative correlation.

3.4.1. Relationship of the Model Performance to Partial APSIM Output Variables and the Constructed Feature

From the color distribution in Figure 9, we observed three different linear correlations among the outputs of the APSIM-ML prediction models. These correlations were observed with RandomForest, DecisionTree, and ExtraTrees, as well as with LASSO, LinearRegression, GradientBoosting, XGB, and LGBM models.

For the DecisionTree and ExtraTrees prediction models, the correlation coefficients between the APSIM outputs and RRMSE range from 0.18 to −0.16. In the case of the RandomForest prediction model, the correlation coefficients between the features and RRMSE range from 0.33 to −0.38, with no correlation found with the simulated daily root depth feature. The RandomForest model shows strong negative correlations between soil moisture stress and photosynthesis, leaf expansion, and positive correlations with humic acid nitrogen mineralization, above-ground dry matter production, soil water supply, and particle nitrogen demand.

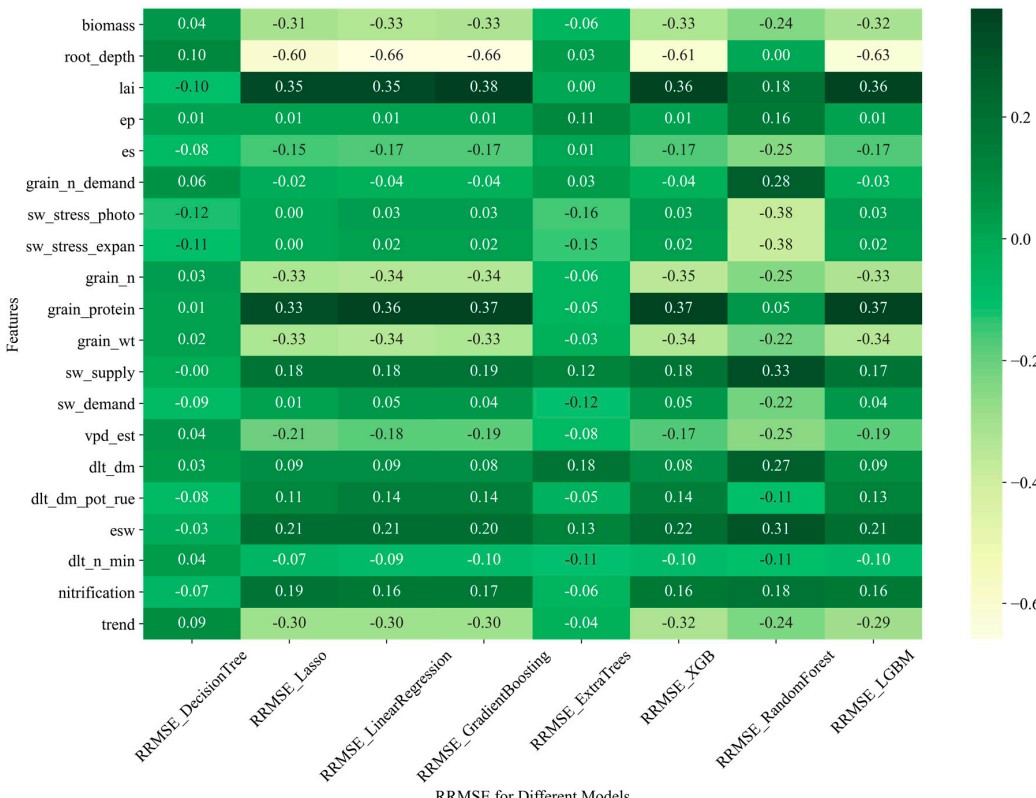

**Figure 9.** The heatmap illustrates the average correlation between the selected variables derived from the APSIM output, constructed variables, and the RRMSE of the various machine learning models across the test years.

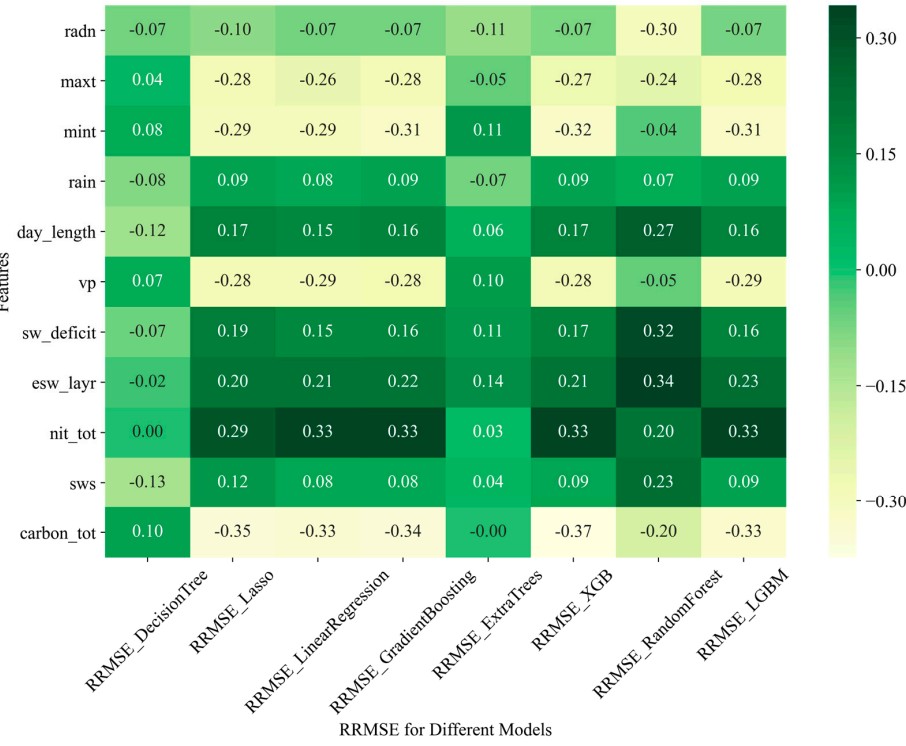

**Figure 10.** Heatmap showing the average correlation between the ML model's RRMSE, weather, and soil characteristics (water, nitrogen, and carbon cycling) across the test years.

In the LASSO, LinearRegression, GradientBoosting, XGB, and LGBM prediction models, the root_depth feature exhibits strong negative correlations with the RRMSE, with correlation coefficients ranging from −0.60 to −0.63. Features like grain_protein and lai show strong positive correlations with the RRMSE, with correlation coefficients ranging from 0.33 to 0.37, while variables like biomass, grain_n, and grain_wt exhibit strong negative correlations.

The constructed feature variable "trend" shows weak correlations with the RRMSE of DecisionTree and ExtraTrees prediction models and strong negative correlations with the other models.

### 3.4.2. Relationship of the Model Performance and Weather Features and the APSIM Output of Soil Variable

Upon examining the color distribution of weather and soil features with respect to the RRMSE across various machine learning prediction models in Figure 10, it is observed that the relationships are similar to those in Figure 9.

In the DecisionTree and ExtraTrees prediction models, the weather and soil features exhibit weak correlations with the model's RRMSE, while they show stronger associations with the accuracy of the RandomForest model. Among the weather features, only sunshine duration (with a correlation coefficient of 0.27) shows a strong positive correlation with the RRMSE of the RandomForest prediction model, while the maximum daily temperature (with a correlation coefficient of −0.24) and radiation intensity (with a correlation coefficient of −0.30) have a significant negative correlation with the model's accuracy.

Among the APSIM output soil features, the RRMSE of the prediction models exhibits a strong negative correlation with the total soil carbon (with a correlation coefficient of −0.20), while showing strong positive correlations with other soil variables in the figure. In the rows of LASSO, LinearRegression, GradientBoosting, XGB, and LGBM regression models, the RRMSE exhibits strong negative correlations with weather features such as the maximum and minimum daily temperatures and atmospheric pressure, with correlation coefficients ranging from −0.26 to −0.31, and negative correlations with radiation intensity, precipitation, and sunshine duration. In the soil features, the total soil carbon shows a strong negative correlation with the model's RRMSE, with correlation coefficients ranging from −0.33 to −0.37, while the total soil nitrogen exhibits a strong positive correlation with the RRMSE, with correlation coefficients ranging from 0.29 to −0.33.

## 4. Discussion

By comparing the performance of each model in these evaluation metrics in Figures 6–8, we found disparities among the predictive models. Specifically, the predictive performance of the weighted average ensemble model aligns with that of the unoptimized weighted ensemble predictive model, while the optimized weighted ensemble model outperforms all unoptimized predictive models. Moreover, APSIM-based machine learning models exhibit a significant advantage over traditional machine learning models in terms of predictive accuracy and interpretability ($R^2$) in specific contexts, likely due to their more accurate modeling of agricultural ecosystems. In ranking the model performances from high to low, we also noticed that, in different test years, some models exhibit an unstable performance. For instance, in the ML models of 2012, the LASSO model's predictive accuracy ranks just below the optimized ensemble model, while in 2016, the predictive performance of LASSO is the lowest. This hints at potential differences in the models' robustness in practical applications. However, the performance of the optimized weighted ensemble model remains consistently high. Ultimately, considering predictive performance, interpretability, and robustness, we believe that the optimized hybrid weighted ensemble model based on APSIM demonstrates the best performance in predicting the wheat yield. These models will offer reliable insights for decision-makers to optimize wheat cultivation and yield forecasting strategies.

Considering the experimental results, the optimized weighted ensemble model exhibited superior performance in wheat yield prediction. Moreover, the predictive model based on APSIM-ML notably outperformed both the ML and APSIM standalone models. However, within the ensemble predictive models optimized using the mean squared error (MSE), the root-mean-square error (RMSE), and the $R^2$, performance variations were observed in terms of the RMSE, MBE, and RRMSE for wheat yield prediction across different scenarios. Consequently, to identify the most suitable predictive model, we conducted an in-depth analysis of the robustness of the predictive models across different test years.

From Figure 11, it is evident that, in 2012, the weighted ensemble model optimized by MSE outperforms the other two models in predicting the wheat yield, especially within the APSIM-ML framework. In 2016, the performance of the model optimized by the RMSE for the weighted ensemble model is slightly superior, but the predictive differences among the three models are not substantial. The MSE-optimized weighted ensemble model within APSIM-ML has an RMSE 2.36 (kg/ha) higher and an MBE −1.34 (kg/ha) lower than the optimal model. In 2021, the interpretability of the models optimized by MSE and $R^2$ for the weighted ensemble is almost identical. The best-performing model in ML is the $R^2$-optimized weighted ensemble, whereas in APSIM-ML, the best is the MSE-optimized weighted ensemble. Notably, the MSE-optimized weighted ensemble maintains a stable level of accuracy across different test years, demonstrating robustness in predicting the wheat yield. Figure 11 indicates that incorporating APSIM-simulated output data significantly enhances the predictive accuracy of ML and APSIM models across test years. Compared to ML, the MBE precision improved by 33.05 (kg/ha), 3.34 (kg/ha), and 9.18 (kg/ha), and compared to APSIM, the MBE precision improved by 106.88 (kg/ha), 99.59 (kg/ha), and 106.18 (kg/ha), respectively.

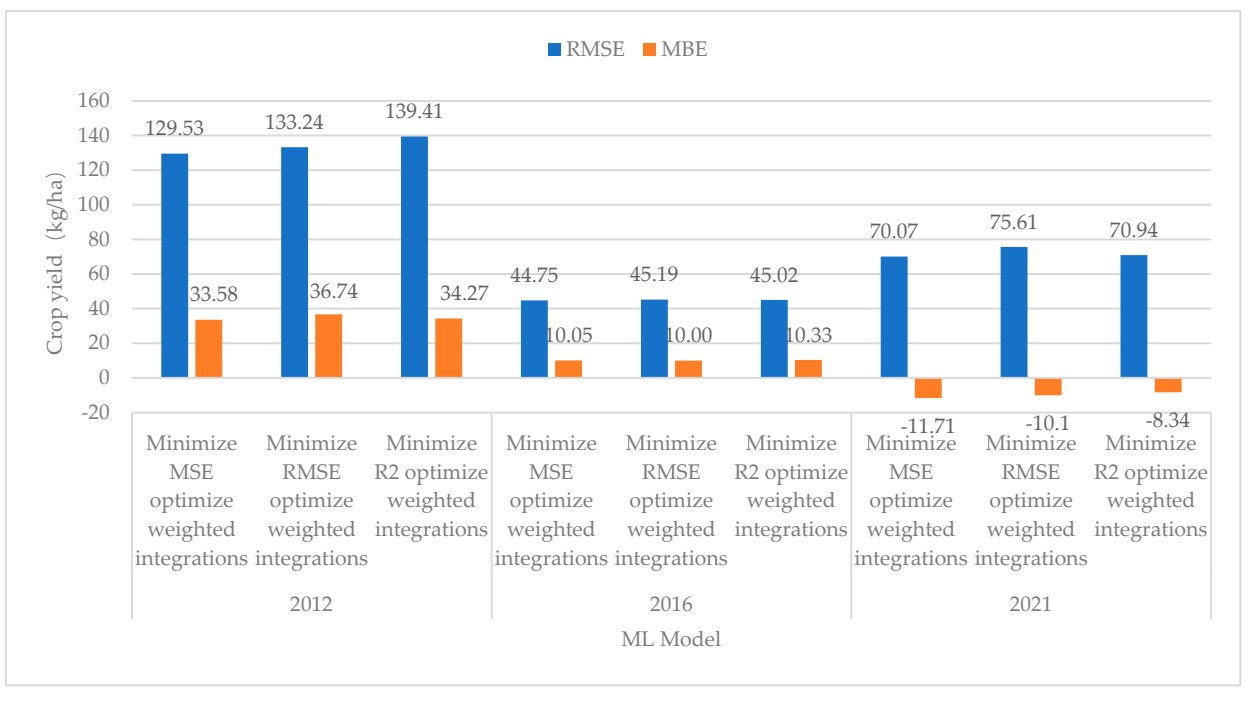

(**a**)

**Figure 11.** *Cont.*

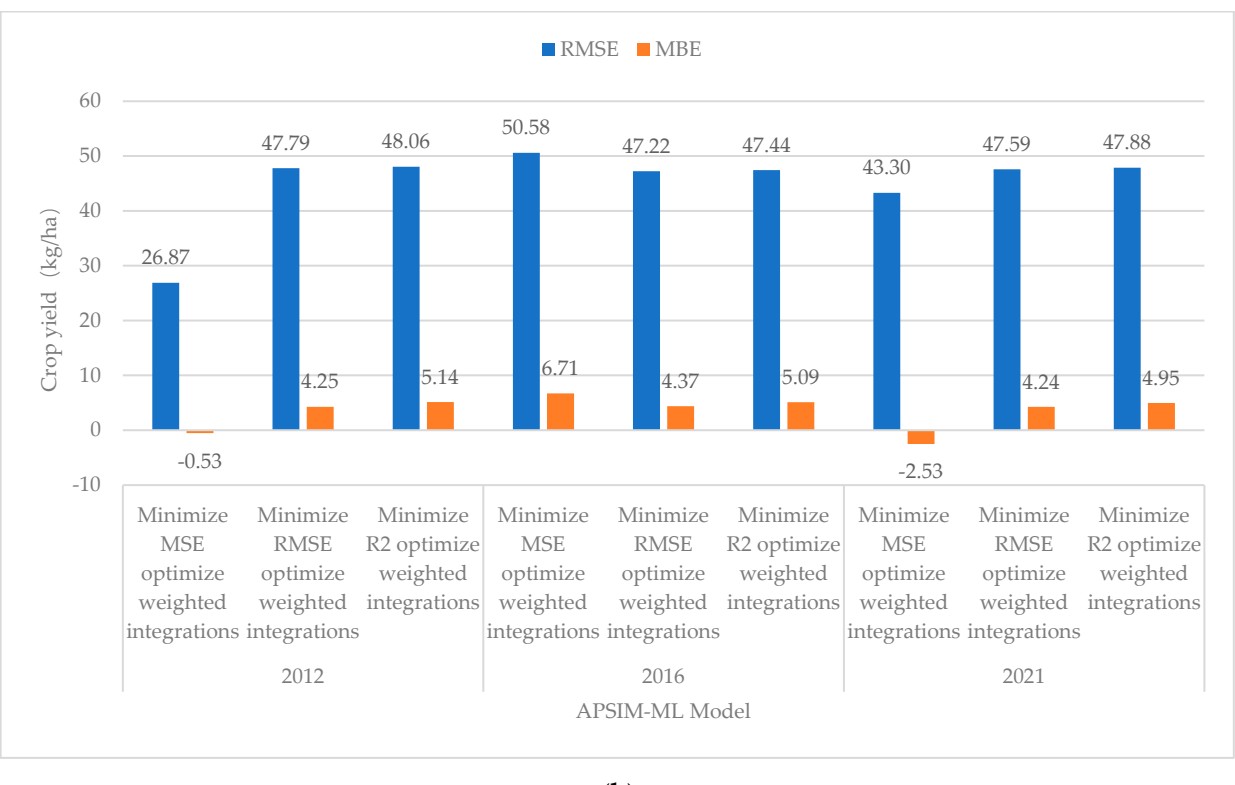

(**b**)

**Figure 11.** The 2012, 2016, and 2021 test data are based on histograms of the three weighted ensemble models, RMSE- and MBE-optimized for ML (**a**) and APSIM-ML (**b**) based on the MSE, RMSE, and $R^2$.

Therefore, we opted for the results derived from the MSE-optimized ensemble model within APSIM-ML as the basis for further analysis (Figure 12). Constructing more robust predictive models for various conditions, including additional meteorological data and incorporating remote sensing or satellite data, could be a potential avenue for future research. Prior studies have also highlighted the effectiveness of weighted optimization with multiple models in providing the best predictions for crop yield, further affirming the efficacy of employing multiple models in agricultural forecasting.

From Figures 9 and 10, it can be observed that there are three types of linear correlations between the eight machine learning prediction models and the input features. The correlation between each prediction model and input features varies in strength, indicating both common and distinct sources of differences in predictive accuracy among different models. Similarly, the sources of the prediction model errors are not limited to one or a few fixed features, but rather result from the complex interactions among features. The relationship between the APSIM-ML model performance and APSIM-simulated output data, soil, and climate input features is clearly evident. These results highlight the critical importance of root_depth for the crop model prediction accuracy, particularly under water stress conditions, where deep-rooted crops may have higher water acquisition capabilities, thereby improving prediction accuracy. Furthermore, the positive correlation between the leaf area index and grain protein confirms the importance of these biological indicators in assessing the crop production potential. The influence of weather features is more complex, with correlations between climate parameters such as sunshine duration, the maximum daily temperature, and radiation revealing the direct impact of climate change on crop growth conditions. In particular, the negative correlation between the soil total carbon and model performance enhancement underscores the potential role of soil management and organic matter improvement in enhancing crop model accuracy. Overall, the significant correlations between the RRMSE and specific weather features and soil water,

nitrogen, and carbon cycling characteristics provide important guidance for accurate crop yield prediction.

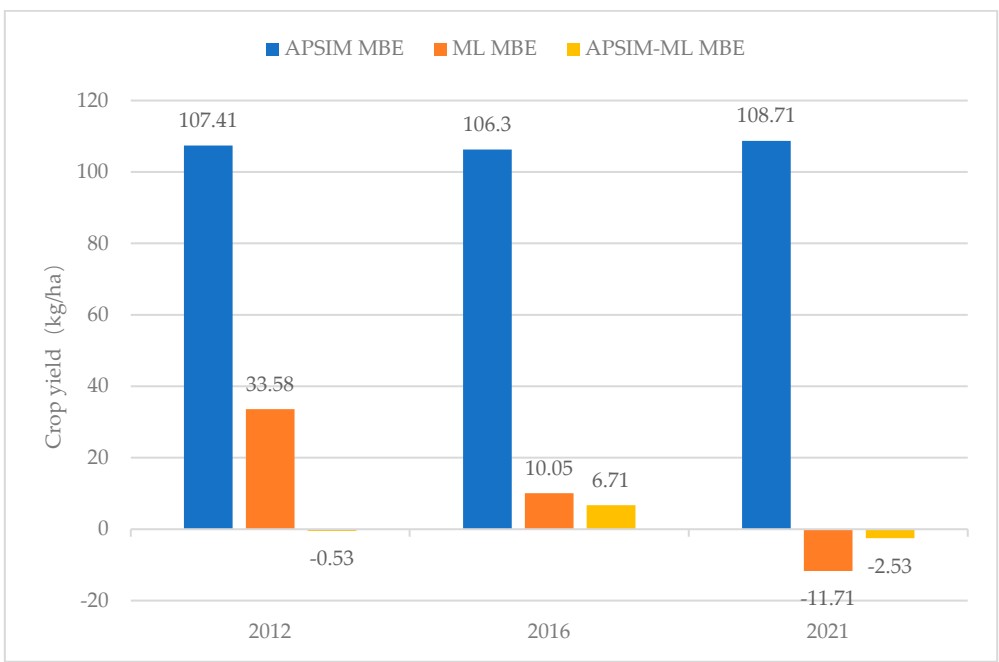

**Figure 12.** MBE comparison chart of the APSIM model, the weighted ensemble model based on ML-minimized MSE optimization, and the APSIM-ML weighted ensemble model for minimizing MSE optimization for predicting wheat yield in 2012, 2016, and 2021.

The weather and soil data have consistently remained crucial input features in crop yield prediction models [80,81]. The weighted ensemble prediction model optimized by minimizing the mean squared error (MSE) demonstrates significant sensitivity to weather variations, including factors such as the daily maximum and minimum temperatures, atmospheric pressure, and solar radiation (Figure 10). The accuracy of the predictive models is significantly influenced by 13 soil parameters derived from nine different soil profiles, including soil bulk density, pH value, organic matter content, soil moisture, available water capacity, and soil hydraulic conductivity. This finding aligns with early research by van Klompenburg et al. [22], who used machine learning to predict the crop yield based on weather and soil information. Similarly, Dai et al. [82] emphasized the importance of soil factors in accurately predicting crop yields in a study conducted in 2011. Recent studies, such as the work by Guo et al. [83] in 2021 and Jiang et al. [84] in the same year, further explore how soil properties and meteorological conditions impact crop yields. Results similar to those proposed by Zou et al. [40], which suggested capturing additional variability in predicting county-level corn yields in China using data from simulated crop model outputs combined with basic learners (RF), were obtained ($R^2 \geq 0.9$, RMSE < 750 kg/ha, and MAE < 500 kg/ha). Both studies confirm the effectiveness of advanced ensemble methods in improving the model prediction accuracy and reducing errors. Unlike the study by Zou et al., we also incorporated crop genetics as input features to construct machine learning (ML) prediction models. These models exhibited particular sensitivity to the feature "trend" constructed from crop genetics and management data. This observation suggests that the trend feature effectively represents the impact of crop genetic evolution on wheat yield.

In addition, after an in-depth analysis using correlation heatmaps, we observed correlations between weather, soil characteristics, APSIM-simulated output features, constructed new features, and model errors. It became evident that there exists some level of correlation between the different prediction models. Interestingly, we found that model errors do not originate from a fixed set of features but result from interactions among various features.

We further conducted an importance analysis on select input features of the optimal model, presenting the average importance in Figure 13. According to the graph, features derived from APSIM-simulated outputs, such as particle nitrogen uptake, particle weight, soil nitrogen content, soil water content, and daily root depth, are the most crucial factors influencing the prediction models. This underscores the significant value of the APSIM simulations in understanding nitrogen and water dynamics between soil and crops for predicting wheat yield.

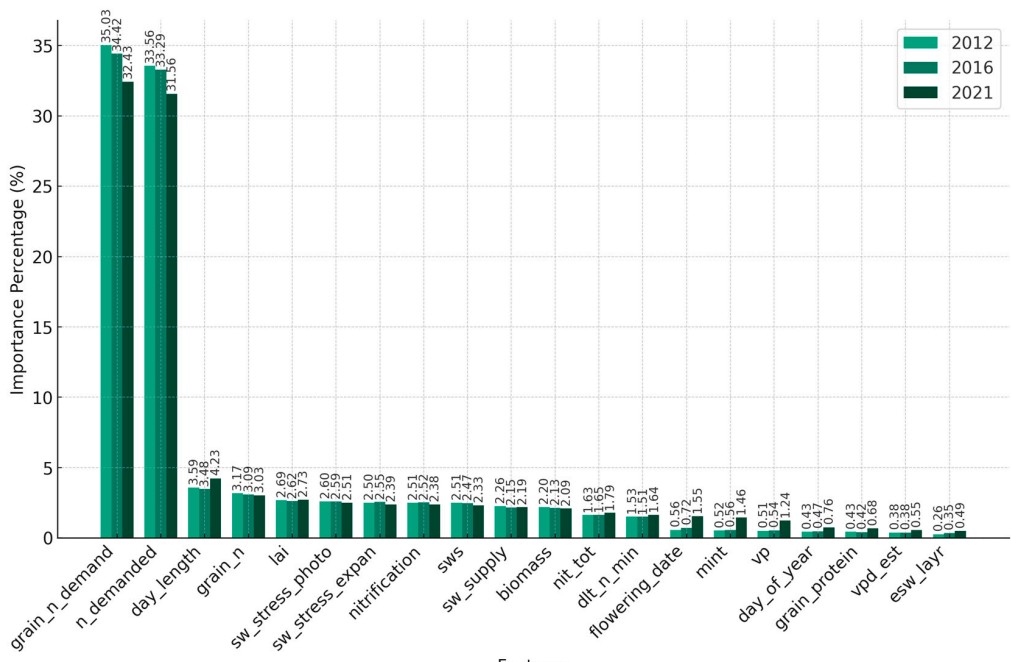

**Figure 13.** The importance of the top 20 input features in the weighted ensemble prediction models optimized to minimize the mean squared error (MSE) in 2012, 2016, and 2020 is evaluated. This evaluation involves multiplying the weight assigned to each base model according to the MSE minimization strategy by the impact of the base model's feature importance and summing them. Finally, the results are presented in percentage form.

In this study, we introduced constructed features alongside APSIM-simulated output data as input features for machine learning models, comparing them against experiments using weather and soil information as machine learning input features. We focused on evaluating the predictive ability of machine learning for wheat yield under different input feature scenarios. Comparing and assessing the correlation between input features and errors in basic models (APSIM and ML) and the coupled model (APSIM-ML), we noted that the APSIM-simulated output features in the prediction models had a greater impact on the results than just weather and soil data. Notably, the simulated output features were derived based on weather and soil information using APSIM simulations. This indicates that, within the modeling process, the interactions between different factors are quite complex and cannot be straightforwardly attributed to specific weather or soil characteristics. Expanding beyond the traditional individual APSIM and ML models, this research applied a coupled model in the wheat cultivation of the loess hilly–gully region in the central–southern part of Gansu Province. The results demonstrate that the ML model coupled with APSIM data outperformed models without APSIM data. Considering the performance, interpretability, and robustness of all predictive models in this comparative study, we conclude that the MSE-optimized hybrid weighted ensemble model based on APSIM performs the best in predicting wheat yield. These models provide reliable references for decision-makers to optimize wheat planting and yield prediction strategies. Consequently, the integration of crop modeling and machine learning stands as an effective approach to enhancing agricultural forecasting accuracy.

While this study has yielded significant findings, there are limitations to consider. For instance, our models are constrained by the dataset; hence, future endeavors might involve introducing a more extensive array of samples and features. Furthermore, exploring more intricate model structures could enhance the predictive accuracy. Through comparisons of various models across different testing years, our study suggests that the minimized MSE optimized weighted ensemble model based on APSIM-ML serves as an effective method for predicting dryland wheat yield, offering robust decision-making support for stakeholders.

## 5. Conclusions

In this study, we significantly improved wheat yield predictions by employing a diverse set of machine learning models combined with crop simulation software (APSIM 7.10). Through comparative analyses, the minimized MSE optimized weighted ensemble model demonstrated outstanding performance across multiple evaluation metrics. It outperformed single models and other ensemble methods, exhibiting more robust and reliable predictions across different test years. The constructed yield trend features and factors such as the flow of water and nitrogen between the soil and crops have a significant impact on the performance of the prediction model. However, when assessing the correlation between the input features and errors across the various machine learning models, we noticed that error sources differed among models, highlighting the complex interplay between input features during modeling. Future research could explore the integration of remote sensing and satellite data, employing APSIM in tandem with deep learning techniques to enhance the wheat yield prediction accuracy and delve deeper into analyzing the factors influencing crop yields.

**Author Contributions:** Conceptualization, Z.L.; methodology, Z.L. and Z.N.; software, Z.L.; validation, Z.L.; formal analysis, Z.L.; investigation, Z.L.; writing—original draft, Z.L.; writing—review and editing, Z.L. and Z.N.; visualization, Z.L.; project administration, Z.L., Z.N. and G.L.; resources, Z.N. and G.L.; supervision, Z.N.; funding acquisition, Z.N. and G.L. All authors have read and agreed to the published version of the manuscript.

**Funding:** This work was supported by the National Natural Science Foundation of China (No. 32160416), Gansu Province Education Department Industrial Support Plan Project (No. 2022CYZC-41), Gansu Agricultural University Youth Mentor Support Fund (No. GAU-QDFC-2022-19), and Gansu Province Top Leading Talent Program (No. GSBJLJ-2023-09).

**Data Availability Statement:** The data presented in this study are available on request from the corresponding author.

**Conflicts of Interest:** The authors declare no conflicts of interest. The funders had no role in the design of the study; in the collection, analyses, or interpretation of data; in the writing of the manuscript; or in the decision to publish the results.

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
