# Peer review of "Integrating Crop Modeling and Machine Learning for the Improved Prediction of Dryland Wheat Yield"

_agronomy, doi:10.3390/agronomy14040777_

Round 1
Reviewer 1 Report
Comments and Suggestions for Authors
General Comments
This study aims to improve the accuracy of predicting dryland wheat yields by integrating crop modeling (APSIM) and machine learning techniques. The subject matter is worthy of investigation and will interest the crop modeling community. It is commendable that the authors used a wide range of data from 1984 to 2021 to train and test their models. I also commend the attempt to discuss the results thoroughly.
Several issues must be addressed in the paper. Importantly, the introduction needs to be restructured for better flow, and the motivation for the study needs to be better articulated. The methodology needs to be summarized where possible, especially the description of the ML algorithms. An important issue about the method of training and validation of the models needs to be addressed. Your findings and results must be compared with the literature findings in the discussion section.
Please find below specific comments and suggestions that need to be addressed.
Specific Comments
1. Line 10-14: This is a long sentence. Please split it for clarity.
2. Line 15: Please define all abbreviations in the abstract e.g. APSIM-ML
3. Line 20: What does this phrase mean? "
provided technical support for the intelligent production of dryland wheat in the loess hilly area. "
4. Line 35: Define abbreviations in their first usage i.e. ML and APSIM.
5. Lines 60-63: Please add a reference to support this statement.
6. Line 63-97: This long paragraph should be split into multiple paragraphs for a better flow of ideas.
7. Line 79-114: A lot of previous research using ML to predict crop yield is presented in the introduction.
While this background is useful, it may be difficult for a reader to glean good information from it. Thus, I suggest the authors create a table summarizing previous studies. The table can have the following columns: study, ML model used, key findings, reference, or whatever columns you come up with to summarize the studies. That will be a cleaner way of displaying these dense paragraphs.
8. Lines 118-124: The motivation for integrating crop models with ML is not strong. You state that meteorological, soil, field management, and crop variety data lack sufficient process-oriented descriptive data about plants and soil. Can the authors elaborate on what they mean by that? Several databases out there contain quality descriptive data that can be fed directly into ML algorithms without first passing through crop models. In other words, why do we need to integrate the two modeling methods when they perform well when used alone?
Please consider refining your motivation for this research.
9. Line 126-127: The authors propose an integrated framework combining APSIM and ML, but this sentence is only the second mention of APSIM in the introduction without describing what it is. Please give some background of the APSIM model in the introduction just like you have done for ML modeling. Additionally, summarize studies that have applied APSIM for crop modeling so the readers can understand what has already been done with APSIM. These studies could even be summarized and included in the table suggested for the ML table.
10. Line 131: The shortcomings in the current models need to be better articulated.
11. Materials and Methods: There is no description of the platform used for the ML modeling. Please include details of the software, packages, versions, etc, so the method is replicable.
12. Line 185: Table 1: Was the soil data collected once or multiple times between the study period i.e. 1984-2021? Please specify
13. Line 225: Can the authors elaborate on how they trained and validated the ML models? For example, was this train/test split, K-fold cross-validation? Etc
14. Table 2: Are these variables present in the APSIM model or were they calculated and output by the APSIM model based on the soil, weather, and management data fed into it?
15. Line 235-245: The mention that data were processed through unit conversion, handing missing data, anomaly removal, normalization, new feature construction, and feature selection. Can the authors please state what was done to the data in each step? For e.g. how many missing data, what anomalies were removed, what new features were added, what features were normalized, what features were selected etc?
16. Line 262-270: It is already known to the reader that missing data is a problem for modeling. Instead of this already-known paragraph, can the authors explain how they handled their data for missing values?
17. Lines 297-310: Here, you are simply telling the reader generic information about what normalizing features mean, which is information the reader probably already knows. Rather, explain how you implemented normalization in your specific data. What and how many features were normalized? What were their ranges?
18. Line 321-323: Where was this domain expert recommendation obtained?
19. Lines 330-333: Was there any procedure followed for this expert-driven feature selection? Removing 1106 features is very significant, therefore, the authors should try to elaborate on how the expert-driven feature selection process was done so it is replicable by other researchers. I wonder why the authors did not use the many available replicable feature selection methods but relied on an expert-driven feature selection process.
20. Line 345: Does the 100 sub-trees represent the 100 features from the previous section?
21. Lines 353-372: This whole section can essentially be boiled down to just this: " To build a high-performing machine learning model, we employed a strategy that combines 20 iterations of Bayesian search with random ten-fold cross-validation to optimize hyperparameter tuning [44]. "
This says exactly what you did without the extraneous sentences.
22. Line 377: What do the authors mean by "intelligently combined"?
23. Lines 397-485: The description of the algorithms is generic information that can be easily found in the literature. Please condense these lines to one paragraph stating what ML algorithms were used and include citations to seminal papers so a reader can go and read more about the algorithm if needed.
Was there a reason why these specific algorithms were chosen, or were they arbitrarily chosen?
24. Line 516: What do you mean by "withstand scrutiny"?
25. Figure 5: The figure caption is not descriptive enough. Yield of what? What years yield data? What does the red-dashed line (45 degree line) mean? How many samples are represented on the plot?
26. Line 522: Please state how the random selection was done.
27. Table 4, 5, and 6: How do have an increase in RRMSE but state negative numbers?
28. Line 565: What do you mean superiority?
29. Figure 6: Please increase the font size of the axes and numbers. They are too small.
30. Lines 610-627: Here, the authors interpret and discuss the results. Thus, this should be moved to the discussion section.
31. Line 614: What do you mean by significant advantage?
32. Line 621-627: The ML models are unstable and not generalizable to new data not used to train them since the performance on the 2012, 2016, and 2021 datasets varies considerably. Therefore, you can’t recommend that the hybrid model be used for predicting wheat yield without a caveat. You could get high or low errors depending on the specific dataset used.
I suggest that the authors combine all the testing data, i.e., 2012+2016+2021, as one before testing unless there is a reason to believe that the conditions (e.g., soil conditions, weather conditions) between the 3 years were significantly different and could have affected the model's performance. In any case, why did the authors not use the k-fold CV method to address the variability in the performance of the three testing datasets? It is a more effective strategy that partitions the dataset into “k” randomly chosen segments, each constituting a certain % of the total data, and conducts the model development and testing cycle “k” times. This method enables the evaluation of models across the entire dataset, offering a more dependable assessment of the regression model's effectiveness.
33. Lines 628-722: The authors mix results with discussion. Please revise only to state your results in the results section and discuss the meaning and implications of those results in the discussion section.
34. Figures 9 and 10: The figure's labels and numbers are unclear. Please increase the font sizes.
35. Line 766: Please support this statement with citation(s).
36. Lines 773-775: It is not enough to mention that your findings align with earlier research. Please discuss your results and findings compared to similar studies in the literature here.
37. Lines 816-818: Can the authors elaborate how their models are constrained by the dataset?
38. Line 819: Intricate model structures like?
Comments on the Quality of English Language
There are some grammatical and spelling issues. The paper can benefit from careful proofreading.
Author Response
Dear Reviewer,
We appreciate the comprehensive review provided by the editors and reviewers, and we are grateful for their insightful comments and suggestions. We have made the corresponding revisions and highlighted them in yellow in the revised manuscript. We believe that these modifications will significantly enhance the academic quality of the paper. Please find detailed responses to the reviewer's comments in the attached document. Once again, we sincerely thank the reviewers for their diligent work and valuable feedback.

Reviewer 2 Report
Comments and Suggestions for Authors
In Section 2 Materials and Methods, page 3, the authors not only describe the used approach but also present some of the results. For example values of MBE and correlation coefficient. In my opinion, that is confusing and should be separated. The results should be described in an appropriate section.
Also in Section 2.3, the authors present some results mixed with a description of the used approach.
The description in Section 2.4 and its subsection is not clear and not detailed. It is not clear what exactly was done and using what methods.
Section 2.4.2 is a generic description of imputation methods but it is not clear what was used by the authors.
In Section 2.4.3 the authors introduce a feature of trend of yield using the linear regression model. In my opinion, when the goal of the research is to predict the yield in the future, special attention should be paid to avoid information leaking when training of models and evaluation of the performance is done. So, for the case when data from a selected year is used as a testing subset, only data from years up to that year (from the past) could be used for model training. The authors should specify, that they used such an approach and that the trend features were calculated, by using only data from the past.
In particular, when the authors used for testing 3 years, I assume that there will be 3 values of trend, calculated accordingly. If the authors used a different approach, that should be revealed to the readers. However, that could limit the reliability of the reported model's performance.
All figures should be corrected. The size of used fonts and the low quality of them make it very difficult to read the figures.
Similar concerns can be raised for the data normalization described in section 2.4.4. What are the values of Xmin, and Xmax in Eq.2 are they calculated using all data (including those from the testing subset), or are each time calculated using a training subset of data (data from the past, which is different for each year used for testing the performance).
Section 2.4.5.1 Does the author present the list of features selected in such a way?
The authors should confirm that the features selection process described in sections 2.4.5.1 and 2.4.5.2 was performed using only the training subset of whole data (not including data from the year used for testing or years after that year). This remark is also related to the necessity of usage of only data from the past and not allowing information leakage, which could lead to an overestimation of the reported model's performance.
The same comment can be raised for hyperparameter tuning described in Section 2.5. Has that operation been performed using only the training data subset and not testing data? To ensure that the obtained prediction performance is fair, there should be no information leakage from the testing dataset to the model-building process.
I have not noticed what software packages were used to perform the machine learning modeling. Maybe it is somewhere hidden in the text, but if so, not easy to find.
In Table 3 and other tables and Figures the authors present the prediction performance obtained by the models for three selected years. As I already noted, it should be clearly explained that the results, for each of year used for testing, were obtained when the model was trained only using data from previous years.
Labels in Fig 13 in axis y are not clear. If I understand this figure correctly, only each second bar has a label, so it is not possible to guess the meaning of the data.
Also, the authors should try to fit all subfigures on one page. Maybe by omitting data with rows of low significance, below some limit, or presenting the top 10 or top 20. Also, common practice is to present the most important data at the top, so the authors should use different sorting of the data on the y-axis.
Author Response
Dear Reviewer,
We appreciate the comprehensive review provided by the editors and reviewers, and we are grateful for their insightful comments and suggestions. We have made the corresponding revisions and highlighted them in blue in the revised manuscript. We believe that these modifications will significantly enhance the academic quality of the paper. Please find detailed responses to the reviewer's comments in the attached document. Once again, we sincerely thank the reviewers for their diligent work and valuable feedback.
